# CROSS-DOMAIN OFFLINE POLICY ADAPTATION WITH DYNAMICS- AND VALUE-ALIGNED DATA FILTERING

## ABSTRACT

Cross-Domain Offline Reinforcement Learning aims to train an agent deployed in the target environment, leveraging both a limited target domain dataset and a source domain dataset with (possibly) sufficient data coverage. Due to the underlying dynamics misalignment between the source and target domain, simply merging the data from two datasets may incur inferior performance. Recent advances address this issue by selectively sharing source domain samples that exhibit dynamics alignment with the target domain. However, these approaches focus solely on dynamics alignment and overlook *value alignment*, i.e., selecting high-quality, high-value samples from the source domain. In this paper, we first demonstrate that both dynamics alignment and value alignment are essential for policy learning, by examining the limitations of the current theoretical framework for cross-domain RL and establishing a concrete sub-optimality gap of a policy trained on the source domain and evaluated on the target domain. Motivated by the theoretical insights, we propose to selectively share those source domain samples with both high dynamics and value alignment and present our **D**ynamics- and **V**alue-aligned **D**ata **F**iltering (DVDF) method. We design a range of dynamics shift settings, including kinematic and morphology shifts, and evaluate DVDF on various tasks and datasets, as well as in challenging extremely low-data settings where the target domain dataset contains only 5,000 transitions. Extensive experiments demonstrate that DVDF consistently outperforms prior strong baselines and delivers exceptional performance across multiple tasks and datasets.

## 1 INTRODUCTION

Reinforcement Learning (RL) (Sutton & Barto, 1999) has recently made remarkable progress in various fields, including video games (Ye et al., 2020; Mnih et al., 2013), robotics (Kober et al., 2013; Kormushev et al., 2013), etc. However, frequent interactions with the environment required for online RL may be expensive, time-consuming, or even risky in real-world applications like healthcare or autonomous navigation. To address this issue, Offline RL (Levine et al., 2020; Prudencio et al., 2023) has been proposed, which aims to learn a well-performing policy in the target environment with access to a pre-collected dataset, thus eliminating the need for interacting with the target environment. Nevertheless, the offline dataset might contain limited data, since collecting sufficient offline data could be costly and even impractical in some scenarios such as medical treatments. The performance of offline RL is often limited by the size of the offline dataset (Levine et al., 2020). If we have access to more datasets, which contain sufficient data collected from environments related to but distinct from the target one (called source environments), we can leverage such additional data for policy learning. This setting is known as Cross-Domain Offline RL (Wen et al., 2024; Liu et al., 2022; 2024a), which aims to train a well-performing agent in the target environment, using both a limited target domain dataset and source domain datasets with wider data coverage.

Although the idea of leveraging additional source domain data to benefit target policy learning is promising, the key challenge lies in that the source and target environment may differ in transition dynamics, and simply merging the source and target data for training could degrade the performance (Wen et al., 2024; Liu et al., 2024a) due to the out-of-distribution (OOD) transition dynamics issue (Liu et al., 2024a). Previous solutions for this issue include training a domain classifier for reward augmentation (Liu et al., 2022; Eysenbach et al., 2020), using supported value optimization and conservative regularization to mitigate overestimation (Liu et al., 2024a), etc. Recent advances (Xu

et al., 2024; Wen et al., 2024; Lyu et al., 2025) introduce dynamics-aware data filtering, where source domain samples are selectively shared based on their alignment with the target dynamics to enhance policy learning. For example, IGDF (Wen et al., 2024) leverages contrastive representation for data filtering, OTDF (Lyu et al., 2025) selects source domain data based on optimal transport. Despite methodological differences, these studies share a common idea: *source domain samples with smaller dynamics misalignment facilitate target policy learning, whereas those with larger misalignment impede it*. However, we argue that this point may not universally hold, as it overlooks the significance of *value alignment*: the selected data should also exhibit high quality other than aligned dynamics. Intuitively, high-quality source samples with moderate dynamics misalignment may contribute more to target policy learning than low-quality samples that are well aligned in dynamics. Consider the case where the source domain dataset consists of non-expert low-quality samples with minor dynamics misalignment and expert samples with larger dynamics misalignment. Methods based on dynamics-aware data filtering will only select low-quality samples, although these samples may contribute little to policy learning. Instead, incorporating expert samples (despite larger dynamics misalignment) may yield better performance. Therefore, we raise the question: *can we devise a cross-domain offline RL method that jointly considers dynamics alignment and value alignment?*

In this paper, we propose a simple yet effective solution for the above question, called **D**ynamics- and **V**alue-aligned **D**ata **F**iltering (DVDF). We start with a motivating example to empirically show that only considering dynamics alignment is not enough for effective cross-domain offline RL. From a theoretical perspective, we reveal that the existing theoretical framework that *tightening the performance discrepancy of a given policy between the source and target domain* **misaligns with the RL objective**, and fails to guarantee learning a well-performing target policy. This explains the limitations of the recent methods like IGDF and OTDF. Alternatively, we derive a concrete sub-optimality bound for policies trained on the source domain and evaluated on the target domain, demonstrating that both dynamics and value alignment are essential for cross-domain offline RL. Based on this theoretical insight, we present our method, DVDF, which utilizes an advantage function pre-trained on the source domain to measure the value misalignment, and incorporates dynamics-aware data filtering to capture the dynamics misalignment within a unified framework. Then DVDF trades off the dynamics and value misalignment and selectively shares source domain samples to train the policy. DVDF can be generally treated as a **plug-in module** and seamlessly integrated with recent methods like IGDF and OTDF. Our contributions can be summarized as follows.

- We examine the limitations of the current theoretical analysis framework for cross-domain offline RL, and theoretically demonstrate that both dynamics alignment and value alignment are essential for cross-domain offline RL, providing new insights for the field.

- Based on the theoretical insight, we propose our method, DVDF, which jointly considers dynamics alignment and value alignment, and selectively shares source domain data for policy learning. DVDF is a plug-in module and can be integrated into other methods like IGDF and OTDF.

- We conduct extensive experiments across various dynamics shift conditions, which demonstrate that DVDF exhibits superior performance on many tasks and datasets compared to strong baselines. We further test DVDF under challenging conditions where the target domain dataset is extremely limited (Lyu et al., 2024b; 2025), and observe DVDF delivers exceptional performance.

## 2 PRELIMINARIES

We consider a Markov Decision Process (MDP) (Puterman, 1990) defined by the six-tuple $\mathcal{M} = (\mathcal{S}, \mathcal{A}, P, r, \rho, \gamma)$ where $\mathcal{S}$ is the state space, $\mathcal{A}$ is the action space, $P : \mathcal{S} \times \mathcal{A} \to \Delta(\mathcal{S})$ is the transition dynamics, $\Delta(\cdot)$ is the probability simplex, $r(s, a) : \mathcal{S} \times \mathcal{A} \to [-r_{\max}, r_{\max}]$ is the reward function, $\rho$ is the initial state distribution, and $\gamma$ is the discount factor. RL aims to learn a policy $\pi : \mathcal{S} \to \Delta(\mathcal{A})$ that maximizes the objective $J_{\mathcal{M}}(\pi) := \mathbb{E}_\pi \left[ \sum_{t=0}^\infty \gamma^t r(s_t, a_t) \right]$.

In the cross-domain RL setting, we assume that we have access to a *source domain* $\mathcal{M}_{\mathrm{src}} = (\mathcal{S}, \mathcal{A}, P_{\mathrm{src}}, r, \rho, \gamma)$ and a *target domain* $\mathcal{M}_{\mathrm{tar}} = (\mathcal{S}, \mathcal{A}, P_{\mathrm{tar}}, r, \rho, \gamma)$. The only difference between the two domains is the transition dynamics. In the offline setting, only a target domain dataset $\mathcal{D}_{\mathrm{tar}} = \{(s_i, a_i, r_i, s_{i+1})\}_{i=1}^{N_1}$ and a source domain dataset $\mathcal{D}_{\mathrm{src}} = \{(s_i, a_i, r_i, s_{i+1})\}_{i=1}^{N_2}$ are available, where $N_1 \ll N_2$. The goal of cross-domain offline RL is to leverage $\mathcal{D}_{\mathrm{tar}}$ and $\mathcal{D}_{\mathrm{src}}$ to improve the performance of the agent in the target domain, where $\mathcal{D}_{\mathrm{src}}$ and $\mathcal{D}_{\mathrm{tar}}$ denote the datasets collected in the source domain and target domain, respectively.

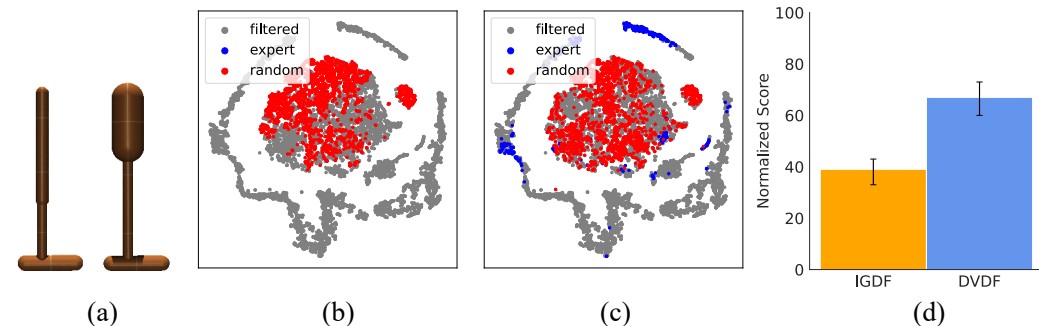

(a)             (b)             (c)             (d)

Figure 1: **(a):** Robot morphology visualization of target domain (left) and source domain (right). **(b):** Source data filtering visualization of IGDF. **(c):** Source data filtering visualization of DVDF. **(d):** Performance comparison between IGDF and DVDF on the target domain.

## 3   MOTIVATING EXAMPLE: DYNAMICS ALIGNMENT ALONE IS INSUFFICIENT

In this section, we use a simple example to demonstrate our claim: solely considering dynamics alignment is insufficient for effective cross-domain offline RL.

We consider the following cross-domain RL scenario: the target domain is `hopper-v2` task from MuJoCo (Todorov et al., 2012), and the source domain is `hopper-v2` task with morphology shift (the head size of the robot is increased), called `hopper-morph-v2`. We visualize the morphology of the robots in Figure 1 (a). In the offline setting, we require both source and target domain datasets. For the target domain, we extract a $10\%$ subset from the `hopper-medium-v2` dataset in D4RL (Fu et al., 2020). The source domain comprises a mixture of: (1) 0.5M random-level samples from `hopper-random-v2` dataset, and (2) 0.5M expert-level samples collected by a well-trained SAC (Haarnoja et al., 2018) policy in the `hopper-morph-v2` environment. Given such source and target domain datasets, we implement the original IGDF (Wen et al., 2024) and our proposed DVDF method (based on IGDF) for source data filtering and policy learning. we set the source data selection ratio to $25\%$ for DVDF and IGDF.

We visualize the source data filtering results of IGDF and DVDF using t-SNE (Van der Maaten & Hinton, 2008), as shown in Figure 1 (b) and (c), respectively. The gray points represent filtered samples, blue points indicate selected expert samples exhibiting dynamics shift, and red points denote selected random samples without dynamics shift. The result reveals that IGDF exclusively selects random samples, whereas DVDF incorporates both random and expert samples. We further evaluate the policies trained by each method in the target environment, with the normalized score presented in Figure 1 (d). The results demonstrate that DVDF achieves a significantly higher average score of **67** compared to 39 obtained by IGDF, representing a **71%** performance improvement. This substantial improvement demonstrates that the shifted expert data can significantly enhance policy learning, validating our motivation that effective cross-domain policy learning requires joint consideration of both dynamics and value alignment. More details can be found in Appendix D.2.

## 4   WHAT IS TRULY ESSENTIAL FOR CROSS-DOMAIN OFFLINE RL?

In this section, we provide theoretical insights for our motivation, by rethinking and examining the limitations of the current theoretical framework for cross-domain RL. Our analysis reveals a fundamental gap in the existing theoretical foundation, prompting us to answer an important question: *what is truly essential for cross-domain offline RL?*

To answer the above question, we first present the key theoretical framework for recent cross-domain RL methods (Wen et al., 2024; Lyu et al., 2025; Xu et al., 2024; Lyu et al., 2024a) in Lemma 4.1, which mainly relies on establishing a performance difference bound of a given policy between the source and target domain:

**Lemma 4.1 (Performance difference bounded by the dynamics misalignment).** *Denote the MDP of the source domain and target domain as $\mathcal{M}_{\mathrm{src}}$ and $\mathcal{M}_{\mathrm{tar}}$. We have the performance difference of*

*a policy $\pi$ under $\mathcal{M}_{\mathrm{src}}$ and $\mathcal{M}_{\mathrm{tar}}$ as below,*

$$|J_{\mathcal{M}_{\mathrm{tar}}}(\pi) - J_{\mathcal{M}_{\mathrm{src}}}(\pi)| \leq C_1 \cdot \underbrace{\sup_{s,a} [D_{\mathrm{TV}}(P_{\mathrm{src}}(\cdot|s,a), P_{\mathrm{tar}}(\cdot|s,a))]}_{\textit{dynamics misalignment}}, \tag{1}$$

*where $C_1 = \frac{2\gamma r_{\max}}{(1-\gamma)^2}$ is a positive constant.*

According to Lemma 4.1, the performance difference is bounded by the dynamics misalignment between the source and target domains. Thus, selectively sharing source domain samples with smaller dynamics misalignment can tighten the performance difference, which builds the theoretical foundation for prior studies (Wen et al., 2024; Xu et al., 2024; Lyu et al., 2024a; 2025). However, a critical limitation of this analysis is that **it misaligns with the RL objective**, that is, obtaining a policy $\pi$ to maximize $J_{\mathcal{M}_{\mathrm{tar}}}(\pi)$. Therefore, tightening such a performance difference bound does not necessarily lead to a well-performing policy in the target domain. Instead, it is more reasonable to narrow the sub-optimality gap of a policy $\pi$ trained on the source domain and evaluated on the target domain. Specifically, we denote the optimal policy in $\mathcal{M}_{\mathrm{src}}$ as $\pi^\star_{\mathrm{src}}$, and the in-sample optimal policy (Kostrikov et al., 2021) extracted from the source domain dataset as $\pi^\star_{\mathrm{insrc}}$. We use $\epsilon_{\mathrm{opt}} := J_{\mathcal{M}_{\mathrm{src}}}(\pi^\star_{\mathrm{src}}) - J_{\mathcal{M}_{\mathrm{src}}}(\pi^\star_{\mathrm{insrc}})$ to represent the inherent performance difference between $\pi^\star_{\mathrm{src}}$ and $\pi^\star_{\mathrm{insrc}}$, and $\epsilon_{\mathrm{opt}}$ is a constant for a given source domain dataset. Our aim is to minimize the sub-optimality gap of $\pi$ in the target domain: $\mathrm{SubOpt} := |J_{\mathcal{M}_{\mathrm{tar}}}(\pi) - J_{\mathcal{M}_{\mathrm{tar}}}(\pi^\star_{\mathrm{tar}})|$, where $\pi^\star_{\mathrm{tar}}$ is the optimal policy in the target domain. We derive the upper bound of this sub-optimality gap in Proposition 4.1.

**Proposition 4.1** (**Sub-optimality gap on target domain**). *Denote the MDP of the source domain and target domain as $\mathcal{M}_{\mathrm{src}}$ and $\mathcal{M}_{\mathrm{tar}}$. For a policy $\pi$ trained on $\mathcal{M}_{\mathrm{src}}$, the sub-optimality gap of $\pi$ on $\mathcal{M}_{\mathrm{tar}}$ can be bounded as below,*

$$\mathrm{SubOpt} \leq \underbrace{|J_{\mathcal{M}_{\mathrm{src}}}(\pi) - J_{\mathcal{M}_{\mathrm{src}}}(\pi^\star_{\mathrm{insrc}})|}_{\textit{(a) value misalignment}} + C_2 \cdot \underbrace{\sup_{s,a} [D_{\mathrm{TV}}(P_{\mathrm{src}}(\cdot|s,a), P_{\mathrm{tar}}(\cdot|s,a))]}_{\textit{(b) dynamics misalignment}} + \epsilon_{\mathrm{opt}}, \tag{2}$$

*where $C_2 = \frac{(2\gamma+2)r_{\max}}{(1-\gamma)^2}$ is a positive constant.*

According to Proposition 4.1, the sub-optimality gap in the target domain can be controlled by two terms: (a) *value misalignment*, representing the sub-optimality on the source domain; (b) *dynamics misalignment* as considered in previous works. To tighten such a sub-optimality gap, both dynamics and value misalignment need to be considered. Hence, we can answer the previous question: *both dynamics and value alignment are essential for effective cross-domain offline RL.*

## 5 DYNAMICS- AND VALUE-ALIGNED DATA FILTERING

Proposition 4.1 conveys a promising way to learn a well-performing policy in the target domain: (1) minimize the dynamics misalignment between the source domain and target domain; (2) minimize the value misalignment between the learned policy and the in-sample optimal policy in the source domain. Neglecting either factor would compromise policy performance. To address this, we adopt a data filtering paradigm inspired by prior works (Xu et al., 2024; Wen et al., 2024; Lyu et al., 2025), which retains source domain data that exhibits aligned dynamics with the target domain, and on which the learned policy can be close to the in-sample optimal policy in the source domain.

### 5.1 ADVANTAGE-AWARE VALUE ALIGNMENT

Given that several existing methods can be applied for measuring dynamics misalignment, such as contrastive learning (Wen et al., 2024) and optimal transport (Lyu et al., 2025), the crucial part remains how to capture value misalignment. To tackle this problem, we derive a lower bound of the value misalignment, which gives us an insight for the practical solution.

**Proposition 5.1** (**Value Misalignment**). *Denote the MDP of the source domain and target domain as $\mathcal{M}_{\mathrm{src}}$ and $\mathcal{M}_{\mathrm{tar}}$, and the behavior policy of the $\mathcal{D}_{\mathrm{src}}$ as $\mu$, for policy $\pi$ trained on $\mathcal{D}_{\mathrm{src}}$, we assume that $(\pi(a|s) - \mu(a|s)) A_\mu(s,a) \geq 0$. Then the value misalignment in Proposition 4.1 can be lower bounded as follows,*

$$J_{\mathcal{M}_{\mathrm{src}}}(\pi) - J_{\mathcal{M}_{\mathrm{src}}}(\pi^\star_{\mathrm{insrc}}) \geq \mathbb{E}_{s\sim\rho_\mu(\cdot), a\sim\mu(\cdot|s)} \left[ A_{\pi^\star_{\mathrm{insrc}}}(s,a) \right] - \mathcal{O}\left( \frac{1}{(1-\gamma)^2} \right), \tag{3}$$

where $A(s, a)$ is the advantage function, and $\rho_\mu(\cdot)$ is the state visiting distribution under $\mu$.

**Remark 1.** *The assumption $(\pi(a|s) - \mu(a|s)) A_\mu(s, a) \geq 0$ indicates that, for any state-action pair $(s, a)$, if $A_\mu(s, a) \geq 0$, i.e., the action $a$ shows superiority over average, the policy $\pi$ has a higher probability than $\mu$ to choose action $a$, and vice versa. Theoretically, this assumption guarantees that the learned policy $\pi$ enjoys a better performance than the behavior policy $\mu$ on the source domain (Proposition 4.1 in (Liu et al., 2024b)). This is reasonable since the goal of offline RL is to outperform the behavior policy. Furthermore, we argue that this assumption could be easily met. If we use IQL (Kostrikov et al., 2021) to optimize the policy $\pi$, since IQL utilizes exponential advantage weighted imitation learning, i.e.,*

$$\pi_{k+1} = \arg\max_{\pi \in \Pi} \mathbb{E}_{(s,a) \in \mathcal{D}} \left[ \exp(\alpha \cdot A_{\pi_k}(s, a)) \log \pi(a|s) \right], \tag{4}$$

*then if the learned policy $\pi$ is initialized as $\mu$, after one step of policy update, any $(s, a)$ with $A_\mu(s, a) > 0$ will be given more weight for imitation, and vice versa. This naturally satisfies our assumption. Therefore, our assumption is quite reasonable and easy to meet.*

The right-hand side in Equation 3 consists of (1) the optimal advantage value on the source domain under the behavior policy's state-action distribution, and (2) a bounded term. Proposition 5.1 gives an important insight that, the value misalignment can be lower bounded by the advantage value under source domain offline data, estimated by the in-sample optimal advantage function on source domain. Given that $J_{\mathcal{M}_{\text{src}}}(\pi) - J_{\mathcal{M}_{\text{src}}}(\pi_{\text{insrc}}^\star) \leq 0$ typically holds, if we want to minimize the value misalignment, we need to maximize $\mathbb{E}_{s \sim \rho_\mu(\cdot), a \sim \mu(\cdot|s)} \left[ A_{\pi_{\text{insrc}}^\star}(s, a) \right]$. This motivates our use of the in-sample optimal advantage function as a quantitative measure for value misalignment, where higher advantage values correspond to lower degrees of value misalignment.

**Remark 2.** *It is worth noting that VGDF (Xu et al., 2024) also emphasizes the importance of value alignment for cross-domain RL and leverages the value function to guide data filtering. However, DVDF fundamentally differs from VGDF in how it interprets value alignment. VGDF defines value alignment as minimizing $|V(s'_{\text{src}}) - V(s'_{\text{tar}})|$, which quantifies dynamics discrepancy from a value difference perspective. That is, **VGDF still only addresses dynamics mismatch**. In contrast, DVDF minimizes $|J_{\mathcal{M}_{\text{src}}}(\pi) - J_{\mathcal{M}_{\text{src}}}(\pi_{\text{insrc}}^\star)|$ for value alignment, which measures policy optimality in the source domain, orthogonal to the dynamics mismatch. Thus, DVDF considers both value and dynamics alignment, distinguishing it from VGDF.*

## 5.2 PRACTICAL IMPLEMENTATION

In Section 5.1, we have demonstrated that an in-sample optimal advantage function could be leveraged for capturing value misalignment. The next question is how to obtain such an advantage function. Since we cannot directly acquire the in-sample optimal advantage function, we propose leveraging a pre-trained offline policy trained on the source domain dataset to approximate the in-sample optimal policy, and using its corresponding advantage function to approximate the in-sample optimal advantage function. However, the advantage approximation error is non-negligible and must be minimized. This raises the question of how to perform offline pre-training effectively. We denote the pre-trained policy as $\pi_{\text{pre}}$, and its advantage function as $A_{\text{pre}}$. We also obtain an advantage function during pre-training, which we label $\hat{A}_{\text{pre}}$. Note that $A_{\text{pre}}$ and $\hat{A}_{\text{pre}}$ are typically different due to the additional conservatism introduced in offline RL. We analyze the advantage approximation error in Proposition 5.2, which provides guidance on how to properly conduct offline pre-training.

**Proposition 5.2 (Advantage Approximation Error).** *Given a pre-trained policy $\pi_{\text{pre}}$ and advantage function $\hat{A}_{\text{pre}}(\cdot)$ to approximate the in-sample optimal policy $\pi_{\text{insrc}}^\star$ and in-sample optimal advantage function $A_{\pi_{\text{insrc}}^\star}(\cdot)$, then the advantage approximation error on offline samples generated by the behavior policy $\mu$ is:*

$$\mathbb{E}_{s \sim \rho_\mu(\cdot), a \sim \mu(\cdot|s)} \left[ \hat{A}_{\text{pre}}(s, a) - A_{\pi_{\text{insrc}}^\star}(s, a) \right] = \Delta J_{\mathcal{M}_{\text{src}}}(\pi_{\text{insrc}}^\star, \pi_{\text{pre}}) + \mathbb{E}_{s \sim \rho_\mu(\cdot), a \sim \mu(\cdot|s)} \left[ \Delta(s, a) \right] \tag{5}$$

*where $\Delta J_{\mathcal{M}_{\text{src}}}(\pi_{\text{insrc}}^\star, \pi_{\text{pre}}) = J_{\mathcal{M}_{\text{src}}}(\pi_{\text{insrc}}^\star) - J_{\mathcal{M}_{\text{src}}}(\pi_{\text{pre}})$, and $\Delta(s, a) = \hat{A}_{\text{pre}}(s, a) - A_{\text{pre}}(s, a)$.*

Proposition 5.2 suggests that minimizing the advantage approximation error requires selecting an offline RL algorithm with two properties: (1) strong empirical performance (to minimize

$\Delta J_{\mathcal{M}_{\text{src}}}(\pi^{\star}_{\text{insrc}}, \pi_{\text{pre}}))$, and (2) accurate advantage estimation (such that $\Delta(s, a)$ is minimized). Although IQL is a natural candidate due to its ability to achieve near in-sample optimal performance across diverse benchmarks and its straightforward advantage estimation, its known tendency for $V$-function underestimation (caused by suboptimal actions) (Xu et al., 2023; Chen et al., 2025) may compromise advantage accuracy and consequently mislead data filtering. To address this limitation while maintaining high performance, we instead adopt Sparse Q-Learning (SQL) (Xu et al., 2023) for pre-training. As an in-sample learning algorithm that explicitly enforces policy sparsity, SQL achieves both near in-sample optimal performance and more reliable advantage estimates, thereby better satisfying our dual requirements of algorithmic performance and advantage accuracy.

After pre-training, we obtain a $Q$-function $\hat{Q}_{\text{pre}}(s, a)$ and a $V$-function $\hat{V}_{\text{pre}}(s)$, we directly derive the advantage function as $\hat{A}_{\text{pre}}(s, a) = \hat{Q}_{\text{pre}}(s, a) - \hat{V}_{\text{pre}}(s)$. Then, we can leverage the pre-trained advantage function as an indicator of value misalignment. The next step is to choose the indicator of the dynamics misalignment. We can just follow previous studies, and apply methods such as contrastive learning (Wen et al., 2024) and optimal transport (Lyu et al., 2025). Here, we follow IGDF (Wen et al., 2024) and measure dynamics misalignment via contrastive learning. Specifically, we train a score function $h(s, a, s')$ via the NCE loss:

$$\mathcal{L}_{\text{NCE}} = -\mathbb{E}_{(s, a, s'_{\text{tar}})} \mathbb{E}_{S'_{\text{src}}} \left[ \log \frac{h(s, a, s'_{\text{tar}})}{\sum_{s' \in \{s'_{\text{tar}}\} \cup S'_{\text{src}}} h(s, a, s')} \right]. \tag{6}$$

where $S'_{\text{src}}$ represents next states from the source dataset. Intuitively, $h(s, a, s')$ assigns high scores when $s' \sim P_{\mathcal{M}_{\text{tar}}}(\cdot | s, a)$, and assigns low scores when $(s, a) \in \mathcal{D}_{\text{tar}}$ and $s' \in \mathcal{D}_{\text{src}}$. Hence, $h(s, a, s')$ can reflect whether the dynamics of the transition $(s, a, s')$ aligns with the target domain dynamics.

Based on the pre-trained advantage function $\hat{A}_{\text{pre}}(\cdot)$ and score function $h(\cdot)$, we propose a practical algorithm, termed DVDF (**D**ynamics- and **V**alue-aligned **D**ata **F**iltering), which selectively shares source domain data with smaller dynamics and value misalignment to train a target policy. Specifically, we define a new score function $g(s, a, s')$:

$$g(s, a, s') = \lambda \cdot h(s, a, s') + (1 - \lambda) \cdot \text{Norm}(\hat{A}_{\text{pre}}(s, a)), \tag{7}$$

where $\lambda$ is a tunable hyperparameter, and $\text{Norm}(\cdot)$ is the min-max normalization operator. $g(\cdot)$ balances value and dynamics misalignment through a simple weighted summation strategy. **This design directly aligns with our theoretical results in Proposition 4.1,** which also combines the two terms via a weighted summation. Then, we extract the top $\xi$-quantile of batch source samples for training, and weigh the Temporal-Difference (TD)-error of selected source data using the score function as in (Wen et al., 2024):

$$\mathcal{L}_Q(\theta) = \frac{1}{2} \mathbb{E}_{(s, a, s') \sim \mathcal{D}_{\text{tar}}} \left[ (Q_\theta - \mathcal{T} Q_\theta)^2 \right] + \frac{1}{2} \mathbb{E}_{(s, a, s') \sim \mathcal{D}_{\text{src}}} \left[ w(s, a, s') g(s, a, s') (Q_\theta - \mathcal{T} Q_\theta)^2 \right], \tag{8}$$

where $w(s, a, s') = \mathbb{I}(g(s, a, s') > g_{\xi\%})$ is an indicator function, and $g_{\xi\%}$ means the $\xi$-th quantile of the $g$-values among source domain samples in a batch. The last step is to update the policy via offline RL algorithms such as IQL. Note that DVDF can serve as a plug-in module and can be combined with different cross-domain offline RL algorithms such as IGDF and OTDF (Lyu et al., 2025), yielding DVDF-IGDF and DVDF-OTDF. We present the detailed algorithm procedure of DVDF-IGDF and DVDF-OTDF in Appendix D.3.

## 6 EXPERIMENTS

In this section, we examine the effectiveness of our method by conducting extensive experiments on environments with various dynamics shifts. We compare the performance of DVDF and other baselines in Section 6.1, and show that DVDF achieves effective offline policy adaptation and beats prior strong baselines across varied dynamics shifts and dataset qualities. In Section 6.2 and Section 6.3, we conduct an ablation study and parameter study for better understanding DVDF.

### 6.1 MAIN RESULTS UNDER VARIOUS DYNAMICS SHIFTS

**Tasks and Datasets.** For the types of dynamics shifts, we consider kinematic shifts and morphology shifts in this paper, and the dynamics shifts are applied to four tasks (halfcheetah, hopper,

Table 1: **Performance comparison under kinematic shifts.** half=halfcheetah, hopp=hopper, walk=walker2d, r=random, m=medium, me=medium-expert, mr=medium-replay, e=expert. We report the normalized score evaluated in the target domain after 1M steps of training, and $\pm$ captures the standard deviation across 5 seeds. We **bold** the highest scores for each task.

| Dataset | IQL | BOSA | DARA | IGDF | DVDF-IGDF | OTDF | DVDF-OTDF |
|---|---|---|---|---|---|---|---|
| half-r | 4.9 | 2.2 | 4.7 | **5.4±0.4** | 4.6±0.1 | **2.2±0.2** | 1.7±0.1 |
| half-m | **45.2** | 39.6 | 44.1 | **45.2±0.1** | 45.1±0.2 | 42.2±0.1 | **45.4±0.6** |
| half-mr | 22.1 | **26.3** | 21.6 | 22.9±1.4 | 26.6±2.3 | 15.6±3.1 | **26.8±4.4** |
| half-me | 43.7 | 42.2 | 52.7 | 57.1±8.9 | **66.7±6.3** | 46.7±4.4 | 45.9±3.0 |
| half-e | 49.7 | **84.3** | 47.4 | 47.6±2.1 | 58.8±4.7 | 79.6±3.0 | **88.9±5.6** |
| hopp-r | 4.5 | **40.7** | 3.8 | **13.0±1.9** | 3.3±0.1 | 2.9±0.4 | **12.6±0.1** |
| hopp-m | 48.8 | **71.4** | 48.8 | 54.3±6.6 | 59.1±3.4 | 46.3±3.7 | **67.8±4.1** |
| hopp-mr | 40.2 | 29.5 | 41.6 | 30.0±5.2 | 32.1±0.8 | 26.2±4.4 | **44.7±2.2** |
| hopp-me | 12.5 | 49.6 | 17.0 | 11.6±0.6 | 60.2±5.9 | 58.1±4.9 | **70.2±7.7** |
| hopp-e | 62.6 | 94.8 | 59.1 | 70.1±3.2 | 83.9±5.0 | 97.0±3.3 | **111.8±4.5** |
| walk-r | 4.0 | 2.2 | 5.1 | 5.2±0.3 | **9.8±1.7** | 0.0±0.0 | 0.0±0.0 |
| walk-m | 48.7 | 44.5 | 43.4 | 51.8±2.4 | 69.7±4.4 | 43.0±2.1 | **71.6±5.9** |
| walk-mr | 12.6 | 4.8 | 15.6 | 11.2±1.1 | 22.6±1.8 | 10.7±1.9 | **25.6±2.4** |
| walk-me | 95.4 | 35.1 | 85.3 | 90.6±3.4 | **104.6±5.1** | 63.1±6.6 | 91.6±8.2 |
| walk-e | 90.1 | 41.9 | 85.5 | 93.7±5.8 | **108.0±4.3** | 98.9±2.1 | 106.0±1.2 |
| ant-r | 11.5 | **31.5** | 10.9 | 13.7±1.9 | 15.6±2.2 | 11.6±1.0 | 25.7±3.4 |
| ant-m | 89.9 | 28.4 | **98.9** | 88.0±4.6 | **98.1±5.0** | 86.1±3.7 | 97.1±5.0 |
| ant-mr | 46.8 | 22.0 | 42.1 | **58.2±9.7** | 44.1±7.6 | **39.6±8.1** | 36.8±3.9 |
| ant-me | 106.1 | 102.5 | 104.8 | 112.8±4.0 | **126.6±7.4** | 105.1±3.9 | **117.2±6.1** |
| ant-e | 111.0 | 57.6 | 115.1 | 119.2±5.6 | **125.2±3.9** | **111.6±2.9** | 107.9±4.0 |
| **Total** | 950.3 | 851.1 | 947.5 | 1001.6 | **1164.7** | 986.5 | **1172.3** |

walker2d, ant) from OpenAI Gym (Brockman et al., 2016). The kinematic shifts are realized by reducing the rotation range of some joints, and the morphology shifts are simulated by modifying the size of some limbs. We defer more details for the realization of dynamics shifts to Appendix C.2 and C.3. Since only a limited amount of target data is accessible, we can sample a percentage of data from offline datasets from D4RL (Fu et al., 2020) as the target domain datasets. We set the percentage to 10% in our experiments. For source domain datasets, we collect data in the modified environments, following a similar data collection process as D4RL. Specifically, We collect datasets of five data qualities (random, medium, medium-replay, medium-expert, expert) with an SAC (Haarnoja et al., 2018) agent trained to different levels of performance in the respective environments, and each source domain dataset contains around 1M samples, much more than the target domain datasets. This amounts to a total of **40** source domain datasets and **20** target domain datasets. Note that for each pair of source and target domain dataset, the type of the tasks and dataset quality remain the same, the difference lies in the transition dynamics and the dataset size. We also examine DVDF in extremely low-data settings (Lyu et al., 2025) where the target dataset contains only 5,000 transitions. We defer the results in Appendix E.2.

**Baselines.** We choose the following baselines for comparison: **IQL** (Kostrikov et al., 2021) (which we train directly on the mixture of source domain data and target domain data). **BOSA** (Liu et al., 2024a), **DARA** (Liu et al., 2022), **IGDF** (Wen et al., 2024) and **OTDF** (Lyu et al., 2025). The backbone of IGDF and OTDF is IQL. We exclude VGDF (Xu et al., 2024) as our baseline since it requires an online target environment. More details about these baselines are presented in Appendix D.1. For our method, we implement DVDF-IGDF and DVDF-OTDF for comparison.

**Experimental Results.** We present the comparison results for each method under kinematic shifts in Table 1. Due to space limit, the results under morphology shifts are deferred to Appendix E.1. We report the normalized score in the target domain. Empirical results demonstrate that DVDF consistently enhances the performance of base algorithms (IGDF and OTDF) while outperforming more baselines (IQL, BOSA, and DARA) across diverse tasks and dataset qualities under kinematic shifts. Notably, DVDF-IGDF surpasses IGDF on **16** out of 20 tasks under kinematic shifts, and DVDF-OTDF achieves a higher score than OTDF on **15** out of 20 tasks under kinematic shifts.

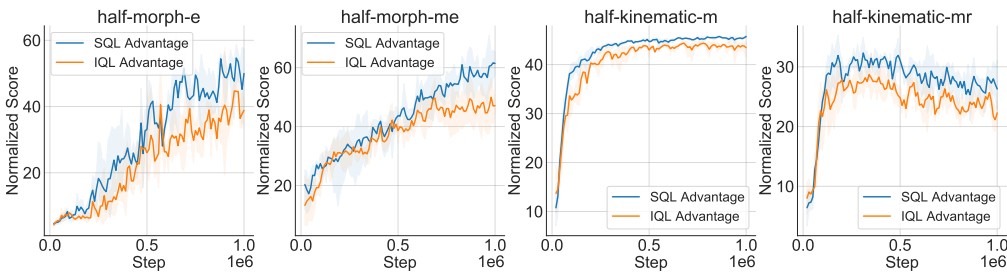

(a) Performance comparison with SQL and IQL pre-trained advantage function.

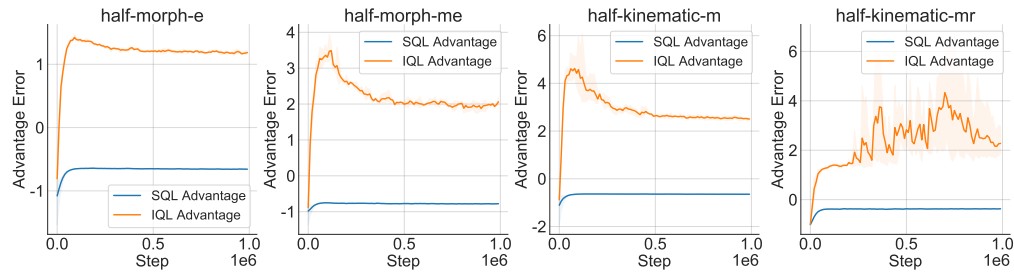

(b) Advantage estimation error comparison with SQL and IQL pre-trained advantage function.

Figure 2: Ablation study on SQL pre-trained advantage function.

Adopting DVDF incurs an increase of **16.3%** (from 1001.6 to 1164.7) and **18.8%** (from 986.5 to 1172.3) to IGDF and OTDF respectively, in terms of the total normalized score under kinematic shifts. Moreover, DVDF shows superiority especially on datasets containing high-quality samples (`medium-expert` and `expert`), outperforming IGDF on **8** out of 8 tasks and OTDF on **6** out of 8 tasks. We attribute this to DVDF's ability to select dynamics- and value-aligned source domain data, whereas IGDF and OTDF may discard substantial value-aligned data.

## 6.2 ABLATION STUDY

In this section, we examine the necessity of SQL pre-training for obtaining the advantage function. We pre-train both SQL and IQL, and apply the resulting advantage functions for data filtering. Experiments are conducted on four datasets using DVDF-IGDF as the base algorithm, with other settings consistent with Section 6.1. We compare the two pre-training methods in terms of algorithm performance and advantage estimation error.

**Performance Comparison.** Figure 2(a) shows the learning curves and performance comparison on four datasets using SQL and IQL pre-trained advantage functions. Clearly, employing the SQL pre-trained advantage function for data filtering yields better performance than using IQL. Therefore, we use the advantage function obtained by SQL pre-training for data filtering in our experiments.

**Advantage Estimation Error Comparison.** We next examine whether the performance improvement from SQL pre-training stems from a more accurate advantage function. We define the advantage estimation error as $\mathcal{E} = \mathbb{E}_{(s,a) \in \mathcal{D}_{\mathrm{src}}} \frac{\hat{A}(s,a) - A(s,a)}{A(s,a)}$, where $\hat{A}(\cdot)$ is the estimated advantage function by either SQL or IQL, and $A(\cdot)$ is the true advantage function obtained by Monte Carlo rollouts. Figure 2(b) shows the advantage estimation error comparison during pre-training. We observe that IQL quickly overestimates the advantage value, consistent with Xu et al. (2023), which notes that IQL tends to underestimate the $V$-function, thereby inflating the estimated advantage. In contrast, SQL typically underestimates the advantage due to the sparsity induced in $V$-function learning. Nevertheless, SQL maintains a smaller estimation error than IQL throughout pre-training, indicating that SQL provides more accurate advantage estimation.

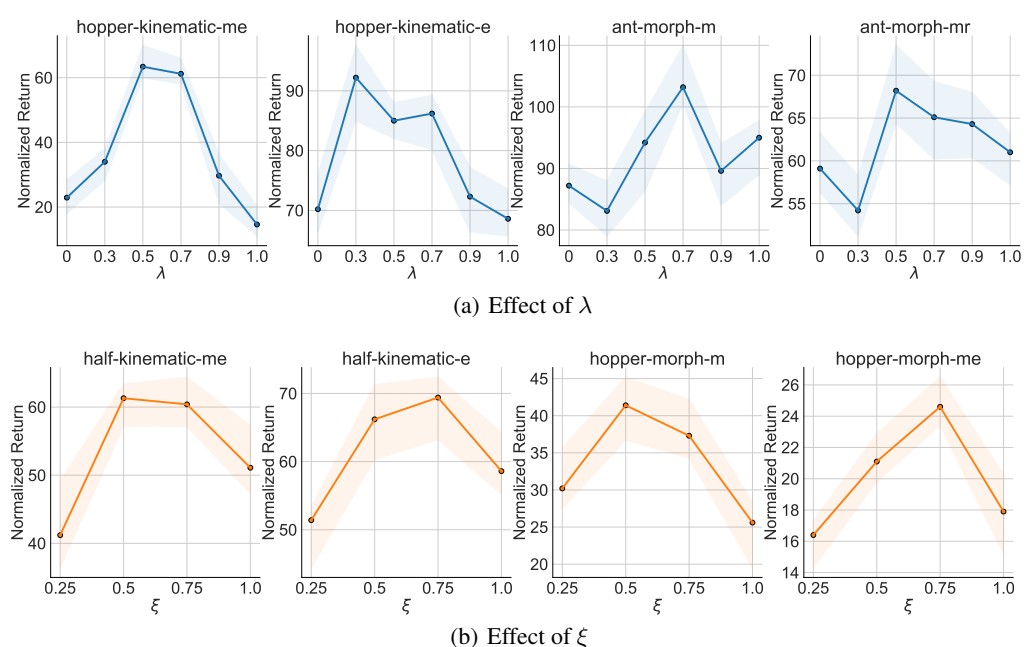

(a) Effect of $\lambda$

(b) Effect of $\xi$

Figure 3: Parameter sensitivity experiments on $\lambda$ and $\xi$.

## 6.3 PARAMETER SENSITIVITY

In this section, we investigate how sensitive DVDF is to the introduced hyperparameters. There are two main hyperparameters in DVDF: the data selection ratio $\xi$ and the alignment tradeoff coefficient $\lambda$. We choose DVDF-IGDF as our base algorithm, running for 1M steps with 10 random seeds to reduce the variance. The dataset setting follows Section 6.1.

**Alignment tradeoff coefficient** $\lambda$**.** The parameter $\lambda$ balances the weight of dynamics alignment and value alignment during data filtering. A larger $\lambda$ emphasizes more on dynamics alignment, and vice versa. We vary $\lambda$ across $\{0.0, 0.3, 0.5, 0.7, 0.9, 1.0\}$ and conduct experiments on four different tasks. Figure 3 (a) shows the effect of different values of $\lambda$ on the final performance, which indicate that neither excessive emphasis on dynamics alignment nor value alignment represents the best choice, and $\lambda = 0.7$ could achieve an effective trade-off between dynamics and value alignment. Therefore, we fix $\lambda = 0.7$ across all datasets in our experiments without further tuning.

**Data selection ratio** $\xi$**.** The parameter $\xi$ decides how many source domain samples can be shared. A larger $\xi$ implies more source domain samples are accepted. To examine its influence, we conduct experiments on four tasks. We sweep $\xi$ across $\{0.25, 0.5, 0.75, 1.0\}$ and present the final performance comparison in Figure 3 (b). We observe an inferior performance when $\xi = 0.25$ or $\xi = 1.0$, and setting $\xi = 0.5$ could achieve a favorable result on most tasks. Therefore, we set $\xi = 0.5$ uniformly for DVDF in all the experiments, instead of performing task-specific tuning.

## 7 RELATED WORK

**Offline RL.** Typical offline RL (Levine et al., 2020; Lange et al., 2012; Prudencio et al., 2023) assumes only access to a static dataset collected in the target environment. Value overestimation may occur in offline RL due to the OOD action issue (Kumar et al., 2020; Fujimoto et al., 2019; Fujimoto & Gu, 2021). Common solutions for this issue include conservative value estimation (Kumar et al., 2020; Lyu et al., 2022; Nikulin et al., 2023; Cheng et al., 2022), adding policy constraints (Kumar et al., 2019; Fujimoto & Gu, 2021; Wu et al., 2021), and augmenting the dataset with dynamics models (Yu et al., 2020; 2021; Kidambi et al., 2020). Our focus is different from these works since we leverage data from another source domain for policy learning.

**Domain Adaptation in RL.** In this work, we investigate the cross-domain policy adaptation problem under dynamics shifts (Xu et al., 2024; Lyu et al., 2024a; Xue et al., 2023), while keeping

other MDP components unchanged. Previous studies for this problem include domain randomization (Slaoui et al., 2019; Mehta et al., 2020), system identification (Clavera et al., 2018; Du et al., 2021), imitation learning (Chae et al., 2022; Kim et al., 2020), and meta RL (Finn et al., 2017; Nagabandi et al., 2018). However, these methods require a manipulable simulator or expert trajectories from the target domain. Recent works (Pan et al.; Guo et al., 2024; 2025; Wang et al., 2024; Niu et al., 2022; 2023) dismiss these limitations and study the setting where limited target domain data and sufficient source domain data are available, either online or offline. Instead, we focus on cross-domain offline RL setting, where both source and target domain data are offline. In this setting, recent studies include reward modification through domain classifier (Liu et al., 2022) or decision transformer (Wang et al., 2024), utilizing supported value optimization (Liu et al., 2024a), leveraging contrastive representation (Wen et al., 2024) or optimal transport (Lyu et al., 2025) for data filtering, and so on. In addition, PSEC (Liu et al., 2025) achieves effective policy adaptation by dynamically composing the parameters of pre-trained source and target domain policies, DmC (Le Pham Van et al., 2025) employs a KNN-based estimator as a measure of dynamics gap, and utilizes the KNN proximity score as a guiding signal for diffusion-based data augmentation. These works have primarily focused on dynamics alignment while neglecting the critical role of value alignment in the source domain. In contrast, DVDF jointly considers dynamics and value alignment, filling this gap in prior research.

## 8 CONCLUSION

In this paper, we investigate cross-domain offline policy adaptation. Through empirical and theoretical analyses, we demonstrate that both dynamics and value alignment are critical for cross-domain offline RL. Building upon this insight, we propose a novel method, DVDF, which leverages a pretrained advantage function to quantify value misalignment and performs data filtering by jointly considering dynamics and value misalignment. Extensive experiments across various dynamics shift scenarios demonstrate that DVDF outperforms prior strong baselines and brings significant performance improvement to base algorithms.

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

## A  EXTENDED RELATED WORKS

**Offline RL.** Offline RL (Levine et al., 2020; Prudencio et al., 2023) suffers from the value overestimation issue (Kumar et al., 2020; Fujimoto et al., 2019) due to the effect of OOD actions (Kumar et al., 2020; 2019; Fujimoto et al., 2019; Fujimoto & Gu, 2021). Model-free offline RL tackles this challenge by restricting the learned policy from producing OOD actions. Common solutions include importance sampling (Gelada & Bellemare, 2019; Liu et al., 2019; Nachum et al., 2019), incorporating policy constraints (Fujimoto & Gu, 2021; Fakoor et al., 2021; Kumar et al., 2019), penalizing value estimation on OOD actions (Kumar et al., 2020; Lyu et al., 2022; Ma et al., 2021; Yang et al., 2022), adopting in-sample learning (Kostrikov et al., 2021; Xu et al., 2023; Garg et al., 2023). However, these methods may induce an over-pessimistic value function, which hinders the generalization of RL agents. Another line of work is model-based offline RL, which enhances the performance and generalization of offline RL by leveraging a dynamics model to generate new samples. Widely used methods include uncertainty estimation (Yu et al., 2020; Kidambi et al., 2020; Sun et al., 2023), learning conservative value functions (Yu et al., 2021; Rigter et al., 2022), representation learning (Lee et al., 2021; Rafailov et al., 2021). All these methods focus on single-domain offline RL, whereas our work investigates cross-domain offline RL, posing additional challenges of dynamics shift.

**Domain Adaptation in RL.** Domain adaptation in RL (Niu et al., 2024) focuses on enhancing the performance of RL agents with the data from the target domain and source domain. In this setting, the agent would inevitably face the challenge of *domain gap*: the discrepancy between the target domain and source domain. This gap encompasses differences in observation (Yang et al., 2023) or action spaces (Zhang et al., 2021; Ge et al., 2023), viewpoints (Liu et al., 2018; Sadeghi et al., 2018), and dynamics (Wen et al., 2024; Lyu et al., 2024a; 2025; Xu et al., 2024; Niu et al., 2022; 2023), among others. In this work, we primarily focus on the problem of cross-dynamics policy adaptation, which means only the transition dynamics differs between the source and target domains. Existing approaches include domain randomization (Slaoui et al., 2019; Mehta et al., 2020), system identification (Clavera et al., 2018; Du et al., 2021), imitation learning (Chae et al., 2022; Kim et al., 2020), and meta RL (Finn et al., 2017; Nagabandi et al., 2018; Li et al., 2020). However, these methods require either a manipulable source domain or expert trajectories in the target domain, which is hard to satisfy in practice. Recent studies (Lyu et al., 2024a; Xu et al., 2024; Niu et al., 2022; 2023) lift the constraints and investigate the setting where limited target domain data or limited interactions with the target domain are available, with abundant data or interactions from the source domain. We study the cross-dynamics offline policy adaptation in this work, where no interactions with the source domain or target domain are permitted, and only limited target domain data and sufficient source domain data with dynamics shifts are available. A crucial problem in this setting is how to measure the dynamics gap between the source and target domains and how to mitigate this dynamics gap. DARA (Liu et al., 2022) trains a domain classifier to estimate the dynamics gap and penalizes the source domain reward by the computed dynamics gap, but the penalty tends to be too pessimistic. IGDF (Wen et al., 2024) proposes a more robust way to estimate the dynamics gap, using the contrastive learning-based mutual information gap as its measure, and adopts a data filtering approach to selectively share the source domain data with a smaller dynamics gap to train the policy. OTDF (Lyu et al., 2025) addresses the extremely limited target domain data setting by estimating the dynamics gap via optimal transport and applying dataset constraints and data filtering to mitigate it. CompFlow (Kong et al., 2025) provides a principled estimation of the dynamics gap via flow matching (Lipman et al., 2022). Other related work includes BOSA (Liu et al., 2024a), which employs supported policy optimization to address the OOD dynamics issue. While these methods primarily focus on dynamics misalignment, our work demonstrates that value misalignment is also critical in cross-domain offline RL, providing a novel insight into the field.

## B  PROOFS OF THEORETICAL RESULTS

In this section, we provide the detailed proofs of the theoretical results in the main text.

## B.1 PROOF OF LEMMA 4.1

*Proof.* The proof starts with

$$
|J_{\mathcal{M}_{\text{tar}}}(\pi) - J_{\mathcal{M}_{\text{src}}}(\pi)| = \left| \frac{\gamma}{1-\gamma} \mathbb{E}_{s,a \sim \rho_{\mathcal{M}_{\text{tar}}}^{\pi}} \left[ \mathbb{E}_{s' \sim P_{\text{tar}}} \left[ V_{\mathcal{M}_{\text{src}}}^{\pi}(s') \right] - \mathbb{E}_{s' \sim P_{\text{src}}} \left[ V_{\mathcal{M}_{\text{src}}}^{\pi}(s') \right] \right] \right|
$$

$$
= \left| \frac{\gamma}{1-\gamma} \mathbb{E}_{s,a \sim \rho_{\mathcal{M}_{\text{tar}}}^{\pi}} \left[ \int_{s'} \left( P_{\text{tar}}(s'|s,a) - P_{\text{src}}(s'|s,a) \right) V_{\mathcal{M}_{\text{src}}}^{\pi}(s') \right] \right|
$$

$$
\leq \frac{\gamma}{1-\gamma} \mathbb{E}_{s,a \sim \rho_{\mathcal{M}_{\text{tar}}}^{\pi}} \left[ \int_{s'} \left| P_{\text{tar}}(s'|s,a) - P_{\text{src}}(s'|s,a) \right| V_{\mathcal{M}_{\text{src}}}^{\pi}(s') \right]
$$

$$
\leq \frac{\gamma \cdot r_{\max}}{(1-\gamma)^2} \mathbb{E}_{s,a \sim \rho_{\mathcal{M}_{\text{tar}}}^{\pi}} \left[ \int_{s'} \left| P_{\text{tar}}(s'|s,a) - P_{\text{src}}(s'|s,a) \right| \right],
$$

where the first equality holds from the telescoping lemma (Xu et al., 2018). Moreover, by the definition of total variance distance, we have

$$
\frac{\gamma \cdot r_{\max}}{(1-\gamma)^2} \mathbb{E}_{s,a \sim \rho_{\mathcal{M}_{\text{tar}}}^{\pi}} \left[ \int_{s'} \left| P_{\text{tar}}(s'|s,a) - P_{\text{src}}(s'|s,a) \right| \right]
$$

$$
= \frac{2\gamma r_{\max}}{(1-\gamma)^2} \mathbb{E}_{s,a \sim \rho_{\mathcal{M}_{\text{tar}}}^{\pi}} \left[ D_{\text{TV}}(P_{\text{tar}}(\cdot|s,a), P_{\text{src}}(\cdot|s,a)) \right]
$$

$$
\leq \frac{2\gamma r_{\max}}{(1-\gamma)^2} \cdot \sup_{s,a} \left[ D_{\text{TV}}(P_{\text{tar}}(\cdot|s,a), P_{\text{src}}(\cdot|s,a)) \right].
$$

Combining the above two inequalities completes the proof. ☐

**Remark.** Let $C_1 = \frac{2\gamma r_{\max}}{(1-\gamma)^2}$, which scales as $\mathcal{O}(\frac{1}{(1-\gamma)^2})$. This constant could be further reduced to $\mathcal{O}(\frac{1}{1-\gamma})$ to ensure a tighter performance bound, under the Lipschitz continuity assumption. Specifically, we introduce the following assumption and corollary.

**Assumption B.1** (Lipschitz Continuity.). *The learned $V$-function is $K_V$-Lipschitz, w.r.t. state $s$, i.e., $\forall s_1, s_2 \in \mathcal{S}, |V(s_1) - V(s_2)| \leq K_V \|s_1 - s_2\|$.*

**Corollary B.1** (Tighter Performance Bound.). *Under Assumption B.1, the performance difference of a policy $\pi$ under $\mathcal{M}_{\text{src}}$ and $\mathcal{M}_{\text{tar}}$ could be more tightly bounded as:*

$$
|J_{\mathcal{M}_{\text{src}}}(\pi) - J_{\mathcal{M}_{\text{tar}}}(\pi)| \leq C \cdot \sup_{s,a} \left[ D_{TV}(P_{\text{src}}(\cdot|s,a), P_{\text{tar}}(\cdot|s,a)) \right], \tag{9}
$$

*where $C = \frac{\gamma}{1-\gamma} \cdot K_V$.*

*Proof.* The conclusion can be directly obtained by following the proof procedure of Theorem 4.5 of (Ji et al., 2022). ☐

## B.2 PROOF OF PROPOSITION 4.1

*Proof.* We first decompose the desired performance bound into four parts:

$$
|J_{\mathcal{M}_{\text{tar}}}(\pi) - J_{\mathcal{M}_{\text{tar}}}(\pi_{\text{tar}}^{\star})|
$$
$$
= | \left( J_{\mathcal{M}_{\text{src}}}(\pi) - J_{\mathcal{M}_{\text{src}}}(\pi_{\text{insrc}}^{\star}) \right) + \left( J_{\mathcal{M}_{\text{tar}}}(\pi) - J_{\mathcal{M}_{\text{src}}}(\pi) \right) + \left( J_{\mathcal{M}_{\text{src}}}(\pi_{\text{src}}^{\star}) - J_{\mathcal{M}_{\text{tar}}}(\pi_{\text{tar}}^{\star}) \right)
$$
$$
+ \left( J_{\mathcal{M}_{\text{src}}}(\pi_{\text{insrc}}^{\star}) - J_{\mathcal{M}_{\text{src}}}(\pi_{\text{src}}^{\star}) \right) |
$$
$$
\leq \underbrace{|J_{\mathcal{M}_{\text{src}}}(\pi) - J_{\mathcal{M}_{\text{src}}}(\pi_{\text{insrc}}^{\star})|}_{(1)} + \underbrace{|J_{\mathcal{M}_{\text{tar}}}(\pi) - J_{\mathcal{M}_{\text{src}}}(\pi)|}_{(2)} + \underbrace{|J_{\mathcal{M}_{\text{src}}}(\pi_{\text{src}}^{\star}) - J_{\mathcal{M}_{\text{tar}}}(\pi_{\text{tar}}^{\star})|}_{(3)} \tag{10}
$$
$$
+ \underbrace{|J_{\mathcal{M}_{\text{src}}}(\pi_{\text{insrc}}^{\star}) - J_{\mathcal{M}_{\text{src}}}(\pi_{\text{src}}^{\star})|}_{(4)}
$$

Part (1) is exactly the desired term (a), i.e., the sub-optimality on the source domain. To get term (b), we need to bound parts (2) and (3), respectively.

We first bound part (2). By directly using Lemma 4.1, we have:

$$(2) := |J_{\mathcal{M}_{\text{tar}}}(\pi) - J_{\mathcal{M}_{\text{src}}}(\pi)|$$

$$\leq \frac{2\gamma r_{\max}}{(1-\gamma)^2} \cdot \sup_{s,a} \left[ D_{\text{TV}}(P_{\text{tar}}(\cdot|s,a), P_{\text{src}}(\cdot|s,a)) \right] \tag{11}$$

Next, we bound part (3), i.e., the performance discrepancy between the optimal policy of two different MDPs.

For part (3), according to the definition of $J_{\mathcal{M}}(\pi)$, we have $J_{\mathcal{M}}(\pi) = V_{\mathcal{M},h=0}^{\pi}(s) := \mathbb{E}_{s \sim \rho_{\mathcal{M}}}[V_{\mathcal{M}}^{\pi}(s)]$. To get the performance bound between two optimal policies in two MDPs, we can turn to calculate the optimal value difference of two MDPs at horizon 0:

$$(3) := \left| V_{\mathcal{M}_{\text{src}},h=0}^{\star}(s) - V_{\mathcal{M}_{\text{tar}},h=0}^{\star}(s) \right| \tag{12}$$

To compute Equation 12, we first consider the value difference at horizon $h-1$:

$$V_{\text{src},h-1}^{\star}(s) - V_{\text{tar},h-1}^{\star}(s)$$

$$= \max_{a \in \mathcal{A}} \int_{s'} P_{\text{src}}(s'|s,a) \left( r(s,a) + \gamma V_{\text{src},h}^{\star}(s') \right) - \max_{a \in \mathcal{A}} \int_{s'} P_{\text{tar}}(s'|s,a) \left( r(s,a) + \gamma V_{\text{tar},h}^{\star}(s') \right)$$

$$= \int_{s'} P_{\text{src}}(s'|s,a_1) \left( r(s,a_1) + \gamma V_{\text{src},h}^{\star}(s') \right) - \int_{s'} P_{\text{tar}}(s'|s,a_2) \left( r(s,a_2) + \gamma V_{\text{tar},h}^{\star}(s') \right)$$

$$\leq \int_{s'} P_{\text{src}}(s'|s,a_1) \left( r(s,a_1) + \gamma V_{\text{src},h}^{\star}(s') \right) - \int_{s'} P_{\text{tar}}(s'|s,a_1) \left( r(s,a_1) + \gamma V_{\text{tar},h}^{\star}(s') \right)$$

$$= \int_{s'} \left( P_{\text{src}}(s'|s,a_1) - P_{\text{tar}}(s'|s,a_1) \right) r(s,a_1) + \gamma \int_{s'} \left( P_{\text{src}}(s'|s,a_1) V_{\text{src},h}^{\star}(s') - P_{\text{tar}}(s'|s,a_1) V_{\text{tar},h}^{\star}(s') \right)$$

$$\leq \max_{a \in \mathcal{A}} \int_{s'} \left( P_{\text{src}}(s'|s,a) - P_{\text{tar}}(s'|s,a) \right) r(s,a) + \max_{a \in \mathcal{A}} \left[ \gamma \int_{s'} \left( P_{\text{src}}(s'|s,a) V_{\text{src},h}^{\star}(s') - P_{\text{tar}}(s'|s,a) V_{\text{tar},h}^{\star}(s') \right) \right]$$

where in the second equality, $a_1 = \arg\max_{a \in \mathcal{A}} \int_{s'} P_{\text{src}}(s'|s,a) \left( r(s,a) + \gamma V_{\text{src},h}^{\star}(s') \right)$, and $a_2 = \arg\max_{a \in \mathcal{A}} \int_{s'} P_{\text{tar}}(s'|s,a) \left( r(s,a) + \gamma V_{\text{tar},h}^{\star}(s') \right)$. In addition, we have

$$\max_{a \in \mathcal{A}} \int_{s'} \left( P_{\text{src}}(s'|s,a) - P_{\text{tar}}(s'|s,a) \right) r(s,a) \leq \max_{a \in \mathcal{A}} \int_{s'} |P_{\text{src}}(s'|s,a) - P_{\text{tar}}(s'|s,a)| \cdot r_{\max},$$

$$\max_{a \in \mathcal{A}} \left[ \gamma \int_{s'} \left( P_{\text{src}}(s'|s,a) V_{\text{src},h}^{\star}(s') - P_{\text{tar}}(s'|s,a) V_{\text{tar},h}^{\star}(s') \right) \right]$$

$$\leq \gamma \max_{a \in \mathcal{A}} \int_{s'} P_{\text{tar}}(s'|s,a) \left( V_{\text{src},h}^{\star}(s') - V_{\text{tar},h}^{\star}(s') \right) + \gamma \max_{a \in \mathcal{A}} \int_{s'} |P_{\text{src}}(s'|s,a) - P_{\text{tar}}(s'|s,a)| V_{\text{src},h}^{\star}(s').$$

Therefore, combining them together, we will obtain

$$V_{\text{src},h-1}^{\star}(s) - V_{\text{tar},h-1}^{\star}(s)$$

$$\leq \frac{2r_{\max}}{1-\gamma} \sup_{s,a} \left[ D_{\text{TV}}(P_{\text{tar}}(\cdot|s,a), P_{\text{src}}(\cdot|s,a)) \right] + \gamma \max_{s' \in \mathcal{S}} \left[ V_{\text{src},h}^{\star}(s') - V_{\text{tar},h}^{\star}(s') \right]. \tag{13}$$

If we denote

$$a_h := \max_{s \in \mathcal{S}} \left[ V_{\text{src},h}^{\star}(s) - V_{\text{tar},h}^{\star}(s) \right] \tag{14}$$

and

$$c := \frac{2r_{\max}}{1-\gamma} \sup_{s,a} \left[ D_{\text{TV}}(P_{\text{tar}}(\cdot|s,a), P_{\text{src}}(\cdot|s,a)) \right], \tag{15}$$

then Equation 13 can be simplified as:

$$a_{h-1} \leq c + \gamma a_h$$

$$\Rightarrow a_{h-1} - \frac{c}{1-\gamma} \leq \gamma \cdot \left( a_h - \frac{c}{1-\gamma} \right) \tag{16}$$

Equation 16 is a recursive expression. To repeat the process recursively, we can easily get:

$$a_0 - \frac{c}{1-\gamma} \le \gamma^H \left( a_H - \frac{c}{1-\gamma} \right) \tag{17}$$

where $H$ denotes the maximum horizon of an episode[1]. According to the definition of the state value, the value of the terminal state is 0, thus $a_H = 0$. Plugging $a_H = 0$ into Equation 17, we can get:

$$V^\star_{\text{src},0}(s) - V^\star_{\text{tar},0}(s) \le a_0 \le c \cdot \frac{1-\gamma^H}{1-\gamma} \tag{18}$$

If we set $H \to \infty$, we have:

$$V^\star_{\text{src},0}(s) - V^\star_{\text{tar},0}(s) \le \frac{2r_{\max}}{(1-\gamma)^2} \sup_{s,a} \left[ D_{\text{TV}}(P_{\text{tar}}(\cdot|s,a), P_{\text{src}}(\cdot|s,a)) \right] \tag{19}$$

Due to the interchangeability of $\mathcal{M}_{\text{src}}$ and $\mathcal{M}_{\text{tar}}$, we also have:

$$V^\star_{\text{src},0}(s) - V^\star_{\text{tar},0}(s) \ge -\frac{2r_{\max}}{(1-\gamma)^2} \sup_{s,a} \left[ D_{\text{TV}}(P_{\text{tar}}(\cdot|s,a), P_{\text{src}}(\cdot|s,a)) \right] \tag{20}$$

Therefore,

$$\begin{aligned} (3) &:= \left| V^\star_{\mathcal{M}_{\text{src}},h=0}(s) - V^\star_{\mathcal{M}_{\text{tar}},h=0}(s) \right| \\ &\le \frac{2r_{\max}}{(1-\gamma)^2} \sup_{s,a} \left[ D_{\text{TV}}(P_{\text{tar}}(\cdot|s,a), P_{\text{src}}(\cdot|s,a)) \right] \end{aligned} \tag{21}$$

Combining the bounds of terms (2) and (3), we get

$$(2) + (3) \le C_2 \cdot \sup_{s,a} \left[ D_{\text{TV}}(P_{\text{tar}}(\cdot|s,a), P_{\text{src}}(\cdot|s,a)) \right] \tag{22}$$

where $C_2 = \frac{(2\gamma+2)r_{\max}}{(1-\gamma)^2}$.

For the term (4), its value is exactly $\epsilon_{\text{opt}}$ by the definition of $\epsilon_{\text{opt}}$. This concludes the proof. $\square$

**Remark.** Similar to Lemma 4.1, $C_2$ could be further reduced to $C = \frac{2\gamma}{1-\gamma} \cdot K_V$ under the Lipschitz continuity assumption.

### B.3 PROOF OF PROPOSITION 5.1

*Proof.* We first divide the sub-optimality on the source domain into two terms:

$$J_{\mathcal{M}_{\text{src}}}(\pi) - J_{\mathcal{M}_{\text{src}}}(\pi^\star_{\text{insrc}}) = \underbrace{J_{\mathcal{M}_{\text{src}}}(\mu) - J_{\mathcal{M}_{\text{src}}}(\pi^\star_{\text{insrc}})}_{(i)} + \underbrace{J_{\mathcal{M}_{\text{src}}}(\pi) - J_{\mathcal{M}_{\text{src}}}(\mu)}_{(ii)} \tag{23}$$

We first compute term (i). By using performance difference lemma (Kakade & Langford, 2002), the return difference between $\pi$ and $\pi^\star_{\text{insrc}}$ in $\mathcal{M}_{\text{src}}$ gives:

$$\begin{aligned} J_{\mathcal{M}_{\text{src}}}(\mu) - J_{\mathcal{M}_{\text{src}}}(\pi^\star_{\text{insrc}}) &= \int_s d_\mu(s) \int_a \left[ \mu(a|s) A_{\pi^\star_{\text{insrc}}}(s,a) \right] \\ &= \mathbb{E}_{s \sim d_\mu(\cdot), a \sim \mu(\cdot|s)} \left[ A_{\pi^\star_{\text{insrc}}}(s,a) \right] \end{aligned} \tag{24}$$

Thus we get the first term in the desired bound. Then we turn to derive the second term.

Based on the Corollary 1 in (Achiam et al., 2017), we have:

$$J_{\mathcal{M}_{\text{src}}}(\pi) - J_{\mathcal{M}_{\text{src}}}(\mu) \ge \int_s d_\mu(s) \int_a \left[ \pi(a|s) A_\mu(s,a) \right] - \frac{2\gamma \epsilon^\pi_\mu}{(1-\gamma)^2} D^{d_\mu}_{TV}(\pi,\mu) \tag{25}$$

---

[1]We focus on the infinite-horizon setting, and $H$ only serves as an intermediate variable for analysis.

where $\epsilon_\mu^\pi = \max_s [\mathbb{E}_{a\sim\pi} A_\mu(s,a)]$, and $D_{TV}^{d_\mu}(\pi,\mu) = \frac{1}{2}\int_s d_\mu(s)\int_a |\pi(a|s) - \mu(a|s)|$ is the total variance distance between $\pi$ and $\mu$ over the distribution $d_\mu$.

Note that under the assumption that for all $(s,a)$, then $(\pi(a|s) - \mu(a|s)) A_\mu(s,a) \geq 0$, we have:

$$
\int_s d_\mu(s) \int_a [\pi(a|s) A_\mu(s,a)]
$$
$$
= \int_s d_\mu(s) \int_a [(\pi(a|s) - \mu(a|s)) A_\mu(s,a)] + \int_s d_\mu(s) \int_a [\mu(a|s) A_\mu(s,a)] \tag{26}
$$
$$
\geq 0 + 0
$$
$$
= 0
$$

Equation 26 uses the fact that $\int_a [\mu(a|s) A_\mu(s,a)] = 0$. An important fact is $\pi$ is updated via offline RL (such as IQL), which imposes implicit or explicit policy constraints on $\pi$. We follow the constraints in IQL and assume:

$$
\max_s (\mathrm{KL}(\mu(\cdot|s), \pi(\cdot|s)), \mathrm{KL}(\pi(\cdot|s), \mu(\cdot|s))) \leq \epsilon \tag{27}
$$

Then we have:

$$
J_{\mathcal{M}_{\mathrm{src}}}(\pi) - J_{\mathcal{M}_{\mathrm{src}}}(\mu) \geq 0 - \frac{2\gamma\epsilon_\mu^\pi}{(1-\gamma)^2} D_{TV}^{d_\mu}(\pi,\mu)
$$
$$
\geq -\frac{2\gamma\epsilon_\mu^\pi}{(1-\gamma)^2} \int_s d_\mu(s) \min\left(\sqrt{\mathrm{KL}(\mu(\cdot|s), \pi(\cdot|s))}, \sqrt{\mathrm{KL}(\pi(\cdot|s), \mu(\cdot|s))}\right)
$$
$$
\geq -\frac{2\gamma\epsilon_\mu^\pi}{(1-\gamma)^2} \int_s d_\mu(s) \max\left(\sqrt{\mathrm{KL}(\mu(\cdot|s), \pi(\cdot|s))}, \sqrt{\mathrm{KL}(\pi(\cdot|s), \mu(\cdot|s))}\right)
$$
$$
\geq -\frac{2\gamma\epsilon_\mu^\pi}{(1-\gamma)^2} \cdot \sqrt{\epsilon}
$$
$$
\tag{28}
$$

The second inequality results from Pinsker inequality (Csiszár & Körner, 2011). Hence, we conclude that for policy $\pi$ learned on the source domain via IQL, it induces a safe policy improvement:

$$
J_{\mathcal{M}_{\mathrm{src}}}(\pi) - J_{\mathcal{M}_{\mathrm{src}}}(\mu) \geq -\mathcal{O}\left(\frac{1}{(1-\gamma)^2}\right) \tag{29}
$$

By combining the result of Equation 24 and Equation 29, we can get the desired performance bound, thus concluding the proof.

$\square$

## B.4 Proof of Proposition 5.2

*Proof.* We first decompose the objective into two terms:

$$
\mathbb{E}_{s\sim\rho_\mu(\cdot), a\sim\mu(\cdot|s)}\left[\hat{A}_{\mathrm{pre}}(s,a) - A_{\pi_{\mathrm{insrc}}^\star}(s,a)\right] = \mathbb{E}_{s\sim\rho_\mu(\cdot), a\sim\mu(\cdot|s)}\left[A_{\mathrm{pre}}(s,a) - A_{\pi_{\mathrm{insrc}}^\star}(s,a)\right]
$$
$$
+ \mathbb{E}_{s\sim\rho_\mu(\cdot), a\sim\mu(\cdot|s)}\left[\hat{A}_{\mathrm{pre}}(s,a) - A_{\mathrm{pre}}(s,a)\right] \tag{30}
$$

For the first term, using the performance difference lemma (Kakade & Langford, 2002), we have:

$$
J_{\mathcal{M}_{\mathrm{src}}}(\mu) - J_{\mathcal{M}_{\mathrm{src}}}(\pi_{\mathrm{pre}}) = \mathbb{E}_{s\sim\rho_\mu(\cdot), a\sim\mu(\cdot|s)}\left[A_{\pi_{\mathrm{pre}}}(s,a)\right] \tag{31}
$$

$$
J_{\mathcal{M}_{\mathrm{src}}}(\mu) - J_{\mathcal{M}_{\mathrm{src}}}(\pi_{\mathrm{insrc}}^\star) = \mathbb{E}_{s\sim\rho_\mu(\cdot), a\sim\mu(\cdot|s)}\left[A_{\pi_{\mathrm{insrc}}^\star}(s,a)\right] \tag{32}
$$

By subtracting Equation 31 with Equation 32, we can get $\Delta J_{\mathcal{M}_{\mathrm{src}}}(\pi_{\mathrm{insrc}}^\star, \pi_{\mathrm{pre}})$. The second term is exactly $\mathbb{E}_{s\sim\rho_\mu(\cdot), a\sim\mu(\cdot|s)}[\Delta(s,a)]$. This concludes the proof.

$\square$

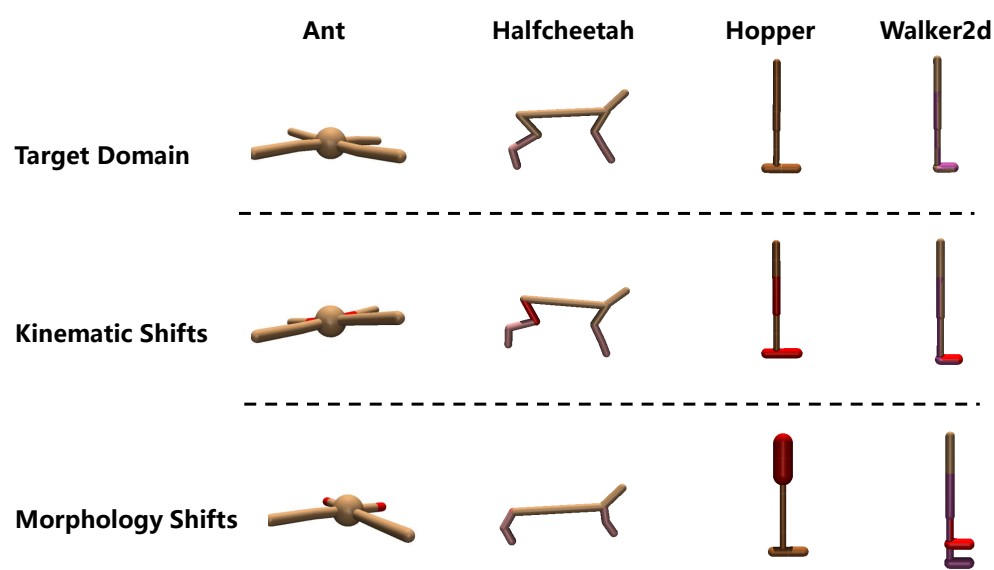

Figure 4: Visualization of the target domains and source domains with kinematic shifts and morphology shifts, across four tasks (`ant`, `halfcheetah`, `hopper`, `walker2d`).

## C  ENVIRONMENT SETTING

In this section, we supplement the detailed environmental settings we adopt in our experiments, including the information of source and target domain datasets, and the code-level realization of kinematic shifts and morphology shifts, etc.

### C.1  DATASETS AND METRICS

**Target domain datasets.** In the cross-domain offline RL setting, the target domain datasets should be collected in the target environment, and only limited target domain data is available. To this end, we directly sample a proportion of data from widely used D4RL (Fu et al., 2020) MuJoCo datasets as the target domain datasets. In D4RL, the MuJoCo datasets are collected with an online SAC (Haarnoja et al., 2018) agent in the environments of Gym (Brockman et al., 2016) simulated by MuJoCo engine (Todorov et al., 2012). We adopt four tasks in our experiments: `halfcheetah-v2`, `hopper-v2`, `walker2d-v2`, `ant-v3`, and consider five dataset qualities for each task: `random`, `medium`, `medium-replay`, `medium-expert`, `expert`. In D4RL, the **random** datasets are collected with a randomly initialized policy. The **medium** datasets contain 1M samples collected from an early-stopped SAC policy. The **medium-replay** datasets record the replay buffer of an SAC agent trained up to the performance of the medium-level agent. The **medium-expert** datasets are a mixture of medium data and expert data at a 50-50 ratio. The **expert** datasets contain 1M samples logged from an expert policy. To construct target domain datasets with different sizes, we could sample different number of data from D4RL datasets. In Section 6.1, we sample 10% data from each D4RL dataset, and in Section E.2, we only sample 5,000 transitions for each dataset to simulate a very limited target data setting.

**Source domain datasets.** To fully examine the effectiveness of our method, we design two kinds of dynamics shift scenarios based on four widely used MuJoCo tasks: `halfcheetah-v2`, `hopper-v2`, `walker2d-v2`, `ant-v3`. The types of dynamics shift we implement include `kinematic shift` and `morphology shift`. The kinematic shift means that some joints of the simulated robot are broken and fail to rotate. The morphology shift indicates that the morphology of the simulated robot in the two domains is different. We show the visualization results of the simulated robot in the source domain and target domain in Figure 4. And the code-level modifications for realizing the dynamics shifts are deferred to the following subsections.

To construct source domain datasets, we follow a similar data-collecting procedure as in D4RL and collect the datasets in the revised environments. We train an online SAC agent within the environments with different kinds of dynamics shifts for 1M steps, and we log the checkpoints of the policy with different training steps and use them to roll out trajectories. The **random** datasets are generated by directly sampling the action space. The **medium** datasets are gathered with the logged policy that exhibits about $1/2$ performance of the expert policy. The **medium-replay** datasets consist of the replay buffer of the medium-level agent. We sample $50\%$ data from the medium datasets and $50\%$ data from the expert datasets, then we mix the sampled data and construct the **medium-expert** datasets. The **expert** datasets are gathered using the last policy checkpoint.

**Metrics.** The metric we use for evaluating the performance of the offline policy in the target domain is the `normalized score` (NS) in D4RL. It is computed as follows:

$$\text{NS} = \frac{J_\pi - J_{\text{random}}}{J_{\text{expert}} - J_{\text{random}}} \times 100\% \tag{33}$$

where $J$ is the return acquired by the agent in the target domain, $J_{\text{random}}$ and $J_{\text{expert}}$ are the returns obtained by the random policy and the expert policy in the target domain, respectively.

## C.2 KINEMATIC SHIFT TASKS

To simulate the kinematic shifts between the source domain and target domain, we modify the `xml` files of the original environments. Specifically, we change the rotation angle of some joints of the simulated robot for different tasks:

***halfcheetah-kinematic***: The rotation angle of the joint on the thigh of the robot's back leg is modified from $[-0.52, 1.05]$ to $[-0.0052, 0.0105]$.

```
# broken back thigh joint
<joint axis="0 1 0" damping="6" name="bthigh" pos="0 0 0" range="
    -.0052 .0105" stiffness="240" type="hinge"/>
```

***hopper-kinematic***: The rotation angle of the head joint is modified from $[-150, 0]$ to $[-0.15, 0]$ and the rotation angle of the foot joint is modified from $[-45, 45]$ to $[-18, 18]$.

```
# broken head joint
<joint axis="0 -1 0" name="thigh_joint" pos="0 0 1.05" range="
    -0.15 0" type="hinge"/>
# broken foot joint
<joint axis="0 -1 0" name="foot_joint" pos="0 0 0.1" range="-18 18
    " type="hinge"/>
```

***walker2d-kinematic***: The rotation angle of the right foot joint is modified from $[-45, 45]$ to $[-0.45, 0.45]$.

```
# broken right foot joint
<joint axis="0 -1 0" name="foot_joint" pos="0 0 0.1" range="-0.45
    0.45" type="hinge"/>
```

***ant-kinematic***: The rotation angles of the joints on the hip of two front legs are modified from $[-30, 30]$ to $[-0.3, 0.3]$.

```
# broken hip joints of front legs
<joint axis="0 0 1" name="hip_1" pos="0.0 0.0 0.0" range="-0.3 0.3
    " type="hinge"/>
<joint axis="0 0 1" name="hip_2" pos="0.0 0.0 0.0" range="-0.3 0.3
    " type="hinge"/>
```

### C.3 MORPHOLOGY SHIFT TASKS

Akin to the kinematic shifts, we modify the morphology of the simulated robot to simulate the morphology shifts:

***halfcheetah-morph***: The sizes of the back thigh and the forward thigh are modified.

```
# back thigh
<geom fromto="0 0 0 -0.0001 0 -0.0001" name="bthigh" size="0.046"
    type="capsule"/>
<body name="bshin" pos="-0.0001 0 -0.0001">
# front thigh
<geom fromto="0 0 0 0.0001 0 0.0001" name="fthigh" size="0.046"
    type="capsule"/>
<body name="fshin" pos="0.0001 0 0.0001">
```

***hopper-morph***: The head size of the robot is modified.

```
# head size
<geom friction="0.9" fromto="0 0 1.45 0 0 1.05" name="torso_geom"
    size="0.125" type="capsule"/>
```

***walker2d-morph***: The thigh on the right leg of the robot is modified.

```
# right leg
<body name="thigh" pos="0 0 1.05">
<joint axis="0 -1 0" name="thigh_joint" pos="0 0 1.05" range="-150
    0" type="hinge"/>
<geom friction="0.9" fromto="0 0 1.05 0 0 1.045" name="thigh_geom"
    size="0.05" type="capsule"/>
<body name="leg" pos="0 0 0.35">
  <joint axis="0 -1 0" name="leg_joint" pos="0 0 1.045" range="
      -150 0" type="hinge"/>
  <geom friction="0.9" fromto="0 0 1.045 0 0 0.3" name="leg_geom"
      size="0.04" type="capsule"/>
  <body name="foot" pos="0.2 0 0">
    <joint axis="0 -1 0" name="foot_joint" pos="0 0 0.3" range="-45
        45" type="hinge"/>
    <geom friction="0.9" fromto="-0.0 0 0.3 0.2 0 0.3" name="
        foot_geom" size="0.06" type="capsule"/>
  </body>
</body>
</body>
```

***ant-morph***: The size of the robot's two front legs is reduced.

```
# front leg 1
<geom fromto="0.0 0.0 0.0 0.1 0.1 0.0" name="left_ankle_geom" size
    ="0.08" type="capsule"/>
# front leg 2
<geom fromto="0.0 0.0 0.0 -0.1 0.1 0.0" name="right_ankle_geom"
    size="0.08" type="capsule"/>
```

## D IMPLEMENTATION DETAILS

In this section, we provide more details about the implementation of the baseline methods and our method. We also list the hyperparameter setup for all methods.

### D.1 BASELINES

**IQL:** IQL (Kostrikov et al., 2021) is an off-the-shelf offline RL algorithm that learns the policy in an *in-sample manner*, which means no OOD samples that lie outside of the offline datasets are required during training. In the cross-domain offline setting, we follow the algorithm procedure but draw samples from both the source domain dataset and the target domain dataset. IQL trains the state value function via expectile regression:

$$\mathcal{L}_V = \mathbb{E}_{(s,a)\sim\mathcal{D}_{\text{src}}\cup\mathcal{D}_{\text{tar}}} \left[ L_2^\tau(Q_{\theta'}(s,a) - V_\psi(s)) \right] \tag{34}$$

where $L_2^\tau(u) = |\tau - \mathbb{I}(u < 0)|\, u^2$, $\mathbb{I}(\cdot)$ is the indicator function, and $\theta'$ is the target network parameter. With such expectile regression, an in-sample optimal value function can be learned. Then the state-action value function is updated by:

$$\mathcal{L}_Q = \mathbb{E}_{(s,a,r,s')\sim\mathcal{D}_{\text{src}}\cup\mathcal{D}_{\text{tar}}} \left[ (r(s,a) + \gamma V_\psi(s') - Q_\theta(s,a))^2 \right] \tag{35}$$

Then the advantage value is calculated as $A(s,a) = Q(s,a) - V(s,a)$ and the policy is extracted by advantage weighted behavior cloning:

$$\mathcal{L}_\pi = -\mathbb{E}_{(s,a)\sim\mathcal{D}_{\text{src}}\cup\mathcal{D}_{\text{tar}}} \left[ \exp(\beta \times A(s,a)) \log \pi_\phi(a|s) \right] \tag{36}$$

where $\beta$ is the inverse temperature coefficient. We implement IQL by following its offlicial code-base[2].

**BOSA:** BOSA (Liu et al., 2024a) defines the issues of the state-action OOD problem and the dynamics OOD problem in cross-domain offline RL, and proposes two support constraints to tackle the issues. To be specific, BOSA handles the OOD state-action problem by supported policy optimization, and mitigates the OOD dynamics problem by supported value optimization. BOSA updates the critic by supported value optimization:

$$\mathcal{L}_Q = \mathbb{E}_{(s,a)\sim\mathcal{D}_{\text{src}}} \left[ Q_{\theta_i}(s,a) \right] + \mathbb{E}_{\substack{(s,a,r,s')\sim\mathcal{D}_{\text{src}}\cup\mathcal{D}_{\text{tar}}, \\ a'\sim\pi_\phi(s')}} \left[ \mathbb{I}(\hat{P}_{\text{tar}}(s'|s,a) > \epsilon)(Q_{\theta_i}(s,a) - y)^2 \right] \tag{37}$$

where $\mathbb{I}(\cdot)$ is the indicator function, and $\hat{P}_{\text{tar}}(s'|s,a)$ is the target domain transition dynamics estimated via maximum likelihood estimation, and $\epsilon$ is the threshold coefficient. The policy in BOSA is updated by supported policy optimization:

$$\mathcal{L}_\pi = \mathbb{E}_{s\sim\mathcal{D}_{\text{src}}\cup\mathcal{D}_{\text{tar}},\, a\sim\pi_\phi(s)} \left[ Q_{\theta_i}(s,a) \right], \quad \text{s.t. } \mathbb{E}_{s\sim\mathcal{D}_{\text{src}}\cup\mathcal{D}_{\text{tar}}} \left[ \hat{\pi}_{\text{mix}}(\pi_\phi(s) \mid s) \right] > \epsilon' \tag{38}$$

where $\epsilon'$ is the threshold coefficient, and $\hat{\pi}_{\phi_{\text{mix}}}(\cdot|s)$ is the empirical behavior policy of the mixed datasets $\mathcal{D}_{\text{src}} \cup \mathcal{D}_{\text{tar}}$ learned with CVAE (Kingma et al., 2013). We do not find the official implementation for BOSA, so we use the codebase[3] in ODRL benchmark (Lyu et al., 2024b), which provides high-quality implementations for various off-dynamics RL algorithms.

**DARA.** DARA (Liu et al., 2022) leverages dynamics-aware reward modification to fulfill dynamics adaptation and is the offline version of DARC (Eysenbach et al., 2020). DARA trains two domain classifiers $q_{\theta_{SAS}}(\text{target}|s_t, a_t, s_{t+1})$ and $q_{\theta_{SA}}(\text{target}|s_t, a_t)$ as follows.

$$\begin{aligned} \mathcal{L}_{\theta_{SAS}} &= \mathbb{E}_{\mathcal{D}_{\text{tar}}} \left[ \log q_{\theta_{SAS}}(\text{target}|s_t, a_t, s_{t+1}) \right] + \mathbb{E}_{\mathcal{D}_{\text{src}}} \left[ \log(1 - q_{\theta_{SAS}}(\text{target}|s_t, a_t, s_{t+1})) \right] \\ \mathcal{L}_{\theta_{SA}} &= \mathbb{E}_{\mathcal{D}_{\text{tar}}} \left[ \log q_{\theta_{SA}}(\text{target}|s_t, a_t) \right] + \mathbb{E}_{\mathcal{D}_{\text{src}}} \left[ \log(1 - q_{\theta_{SA}}(\text{target}|s_t, a_t)) \right] \end{aligned} \tag{39}$$

The two domain classifiers are used to estimate the dynamics gap $\log \frac{P_{\mathcal{M}_{\text{tar}}}(s_{t+1}|s_t, a_t)}{P_{\mathcal{M}_{\text{src}}}(s_{t+1}|s_t, a_t)}$ between the source domain and the target domain. Then the estimated dynamics gap is used as a penalty to the source domain rewards:

$$\hat{r}_{\text{DARA}} = r - \lambda \times \delta_r, \quad \delta_r(s_t, a_t) = -\log \frac{q_{\theta_{\text{SAS}}}(\text{target}|s_t, a_t, s_{t+1}) q_{\theta_{\text{SA}}}(\text{source}|s_t, a_t)}{q_{\theta_{\text{SAS}}}(\text{source}|s_t, a_t, s_{t+1}) q_{\theta_{\text{SA}}}(\text{target}|s_t, a_t)} \tag{40}$$

where $\lambda$ controls the intensity of the reward penalty. We use the re-implementation in ODRL for DARA. $\lambda$ is set to $0.1$, and the reward penalty is clipped within $[-10, 10]$ for training stability.

---

[2]https://github.com/ikostrikov/implicit_q_learning.git
[3]https://github.com/OffDynamicsRL/off-dynamics-rl.git

**IGDF.** IGDF (Wen et al., 2024) estimates the domain gap between the source domain and the target domain with contrastive representation learning, and employs data filtering to share source domain samples with a smaller dynamics gap for training. IGDF trains a score function $h(\cdot)$ using $(s, a, s'_{\text{tar}}) \sim \mathcal{D}_{\text{tar}}$ as the positive samples, and transitions $(s, a, s'_{\text{src}})$ as the negative samples, where $(s, a) \sim \mathcal{D}_{\text{tar}}$ and $s'_{\text{src}} \sim \mathcal{D}_{\text{src}}$. $h(\cdot)$ is trained with the contrastive learning objective:

$$\mathcal{L} = -\mathbb{E}_{(s,a,s'_{\text{tar}})}\mathbb{E}_{s'_{\text{src}}} \left[ \log \frac{h(s, a, s'_{\text{tar}})}{\sum_{s' \in s'_{\text{tar}} \cup s'_{\text{src}}} h(s, a, s')} \right] \tag{41}$$

For the construction of the score function, IGDF adopts two networks $\phi(s, a)$ and $\psi(s')$ to learn the representations of state-action and state, respectively. The score function is expressed as a linear parameterization of $\phi(s, a)$ and $\psi(s')$:

$$h(s, a, s') = \exp(\phi(s, a)^T \psi(s')) \tag{42}$$

Based on the learned score function, IGDF proposes to selectively share source domain data for training value functions:

$$\mathcal{L}_Q = \frac{1}{2}\mathbb{E}_{\mathcal{D}_{\text{tar}}} \left[ (Q_\theta - \mathcal{T}Q_\theta)^2 \right] + \frac{1}{2}\alpha \cdot h(s, a, s')\mathbb{E}_{(s,a,s') \sim \mathcal{D}_{\text{src}}} \left[ \mathbb{I}(h(s, a, s') > h_{\xi\%})(Q_\theta - \mathcal{T}Q_\theta)^2 \right] \tag{43}$$

where $\mathbb{I}(\cdot)$ is the indicator function, $\alpha$ is the weighting coefficient, $\xi$ is the data selection ratio. We implement IGDF by following its official codebase[4].

**OTDF.** OTDF (Lyu et al., 2025) depicts the distance between the source domain data and target domain data by computing the Wasserstein distance (Peyré et al., 2019):

$$\mathcal{W}(u, u') = \min_{\mu \in M} \sum_{t=1}^{|\mathcal{D}_{\text{src}}|} \sum_{t'=1}^{|\mathcal{D}_{\text{tar}}|} C(u_t, u'_{t'}) \cdot \mu_{t,t'} \tag{44}$$

where $u = s_{\text{src}} \oplus a_{\text{src}} \oplus s'_{\text{src}}$, $u' = s_{\text{tar}} \oplus a_{\text{tar}} \oplus s'_{\text{tar}}$, $C$ is the cost function and $M$ is the coupling matrices. After solving Equation 44 for $\mu^\star$, the OTDF determines the deviation between a source domain dataset and the target domain dataset via:

$$d(u_t) = -\sum_{t'=1}^{|\mathcal{D}_{\text{tar}}|} C(u_t, u'_{t'})\mu^\star_{t,t'}, \quad u_t = (s^t_{\text{src}}, a^t_{\text{src}}, (s'_{\text{src}})^t) \sim \mathcal{D}_{\text{src}} \tag{45}$$

Then the critic is updated by:

$$\mathcal{L}_Q = \mathbb{E}_{\mathcal{D}_{\text{tar}}} \left[ (Q_\theta - \mathcal{T}Q_\theta)^2 \right] + \mathbb{E}_{(s,a,s') \sim \mathcal{D}_{\text{src}}} \left[ \exp(\alpha \times d)\mathbb{I}(d > d_{\%})(Q_\theta - \mathcal{T}Q_\theta)^2 \right] \tag{46}$$

Besides, OTDF includes an extra policy regularization term that encourages the policy to be close to the support region of the target dataset:

$$\widehat{\mathcal{L}_\pi} = \mathcal{L}_\pi - \beta \times \mathbb{E}_{s \sim \mathcal{D}_{\text{src}} \cup \mathcal{D}_{\text{tar}}} \log \pi^b_{\text{tar}}(\pi(\cdot|s)|s) \tag{47}$$

where $\mathcal{L}_\pi$ is the original policy optimization objective and $\beta$ is the weight coefficient. We run the official code[5] for OTDF in our experiments.

### D.2 MORE DETAILS OF MOTIVATION EXAMPLE

In this section, we supplement with more details for our motivation example in Section 3.

We provide the visualization results of the random-expert mixed source domain dataset in Figure 5 (a). We further conduct an experiment to demonstrate the necessity of dynamics- and value-aligned data filtering. Instead of using $h(s, a, s')$ or $g(s, a, s')$ as the indicator like IGDF and DVDF, we directly use $A_{\text{pre}}(s, a)$ for data filtering, which means we select source domain data with a smaller value misalignment, disregarding dynamics misalignment. We term this modified algorithm version as Value-IGDF. We visualize the data filtering results (data selection ratio $\xi$ is 25%) in Figure 5 (b), which indicates that the selected samples are predominantly expert samples, despite the dynamics shifts. We evaluate the performance of IGDF, Value-IGDF and DVDF on the source domain and present the results in Figure 5 (c). We can see that while Value-IGDF outperforms IGDF, it still lags behind DVDF, highlighting the necessity of jointly considering dynamics and value alignment.

---

[4]https://github.com/BattleWen/IGDF.git
[5]https://github.com/dmksjfl/OTDF.git

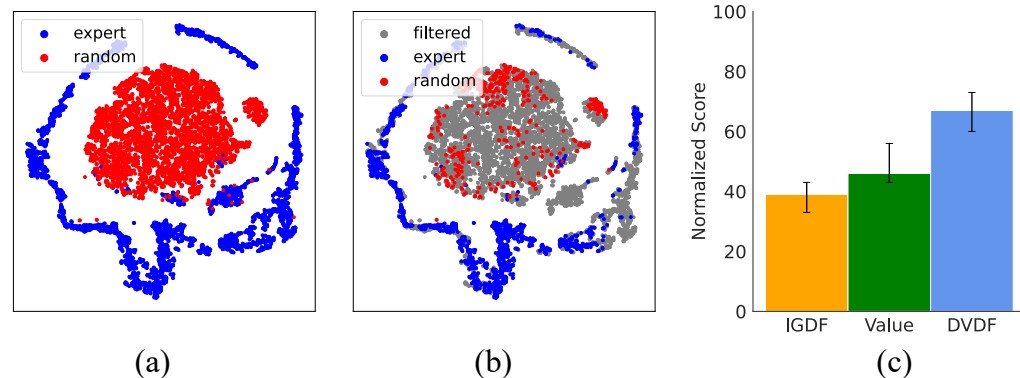

Figure 5: **(a):** Visualization of source domain data. **(b):** Source domain data filtering visualization of Value-IGDF. **(c):** Performance comparison between IGDF, Value-IGDF and DVDF.

### D.3 ALGORITHMIC DETAILS OF DVDF

As a plug-in module, DVDF can be seamlessly integrated into various cross-domain offline RL algorithms, such as IGDF and OTDF. In this section, we present more details about the combination of DVDF with IGDF (tagged DVDF-IGDF) and OTDF (namely DVDF-OTDF), and summarize the pseudocodes of DVDF-IGDF and DVDF-OTDF.

For DVDF-IGDF, the new score function $g(\cdot)$ incorporates the vanilla score function $h(\cdot)$ and the pre-trained advantage function $A_{\text{pre}}(\cdot)$. During training, only $h(\cdot)$ is updated and $A_{\text{pre}}(\cdot)$ remains frozen. We leverage $g(\cdot)$ for data filtering, and the other procedure of DVDF-IGDF keeps identical with that of IGDF. The detailed pseudocode of DVDF-IGDF is presented in Algorithm 1. The blue texts mark the different algorithm procedure from the original IGDF.

For DVDF-OTDF, other than $A_{\text{pre}}(\cdot)$, the optimal coupling $\mu^\star$ also needs to be computed before the policy training process begins. We choose cosine distance as the cost function and utilize OTT-JAX library (Cuturi et al., 2022) for solving the OT problem. Note that DVDF only plays a part in the procedure of data filtering, and the remaining process of training remains the same as that in OTDF. We summarize the pseudocode of DVDF-OTDF in Algorithm 2. The blue lines highlight the different procedure from the vanilla OTDF.

### D.4 HYPERPARAMETER SETUP

We present the main hyperparameter setup in our experiments for all the methods we use in Table 2.

## E WIDER EXPERIMENTAL RESULTS

### E.1 RESULTS UNDER MORPHOLOGY SHIFTS

In Section 6.1, we give the evaluation results of our methods with different size of target domain datasets under kinematic shifts. In this section, we supplement with more evaluation results under morphology shifts, with other experimental settings identical with Section 6.1.

Table 3 presents the comparison results using 10% data of the D4RL datasets as the target domain data. The results clearly show that DVDF enhances the performance of the base algorithms. Specifically, DVDF-IGDF achieves the highest total score among all the methods, outperforming IGDF by **15.4%** (1039.0 to 1198.7) and performing better in **16** out of 20 tasks, while DVDF-OTDF improves OTDF by **11.0%** (1042.1 to 1156.3) and excels in **14** out of 20 tasks.

### E.2 EXTENDED RESULTS WITH EXTREMELY LIMITED TARGET DATA

In this section, we consider a more challenging setting compared with Section 6.1 following (Lyu et al., 2025; 2024b), where only extremely limited target domain data (around 5,000 transitions)

---

**Algorithm 1** DVDF-IGDF

---

1: **Require:** Source domain offline dataset $\mathcal{D}_{\text{src}}$, target domain offline dataset $\mathcal{D}_{\text{tar}}$, mixed offline dataset $\mathcal{D}_{\text{mix}}$

2: **Initialization:** Policy network $\pi_\eta$, value network $V_\beta$, target Q network $Q_\theta$, encoder networks $\phi(s, a)$, $\psi(s')$, data selection ratio $\xi$, batch size $B$, importance coefficient $\alpha$, alignment tradeoff coefficient $\lambda$

3: **// Pre-train the advantage function**

4: Pre-train an SQL agent on $\mathcal{D}_{\text{src}}$, obtain $\hat{A}_{\text{pre}}(\cdot)$ and normalize $\hat{A}_{\text{pre}}(\cdot)$

5: **// Contrastive Representation Learning**

6: Train $h(s, a, s')$ via contrastive representation learning by Equation 41, obtain the score function $\omega(s, a, s') = \lambda \cdot h(s, a, s') + (1 - \lambda) \cdot \hat{A}_{\text{pre}}(s, a)$

7: **// TD Learning**

8: **for** each gradient step **do**

9:     Sample $b_{\text{src}} := \{(s, a, r, s')\}$ from $\mathcal{D}_{\text{src}}$

10:     Sample $b_{\text{tar}} := \{(s, a, r, s')\}$ from $\mathcal{D}_{\text{tar}}$

11:     Sample the top-$\xi$ samples from $b_{\text{src}}$ ranked by $g(s_{\text{src}}, a_{\text{src}}, s'_{\text{src}})$

12:     Compute weights $\omega(s, a, s')$ following:

13:       $\omega(s, a, s') = \mathbb{I}(g(s, a, s') \geq g_{\xi\%})$

14:     **// Optimize the $V_\beta$ function**

15:     Compute loss $\mathcal{L}_V$:

16:       $\mathcal{L}_V = \mathbb{E}_{(s,a)\sim\mathcal{D}_{\text{src}}\cup\mathcal{D}_{\text{tar}}}\left[L_2^\tau\left(Q_\theta(s, a) - V_\beta(s)\right)\right]$

17:     Update $V_\beta$ using $\mathcal{L}_V$

18:     **// Optimize the $Q_\theta$ function**

19:     Compute loss $\mathcal{L}_Q$:

20:       $\mathcal{L}_Q = \frac{1}{2} \cdot \mathbb{E}_{(s,a,r,s')\sim\mathcal{D}_{\text{tar}}}\left[\left(Q_\theta(s, a) - (r + \gamma V_\beta(s'))\right)^2\right]$

21:       $+ \frac{1}{2} \cdot \mathbb{E}_{(s,a,r,s')\sim\mathcal{D}_{\text{src}}}\left[\omega(s, a, s')g(s, a, s')\left(Q_\theta(s, a) - (r + \gamma V_\beta(s'))\right)^2\right]$

22:     Update $Q_\theta$ using $\mathcal{L}_Q$

23:     **// Update target network**

24:     Update target network parameters: $\theta' \leftarrow (1 - \mu)\theta + \mu\theta'$

25:     **// Policy Extraction (AWR)**

26:     Compute advantage $A(s, a) = Q_\theta(s, a) - V_\beta(s)$

27:     Optimize policy network $\pi_\eta$ using advantage-weighted regression (AWR):

28:       $\mathcal{L}_\pi = \mathbb{E}_{(s,a)\sim\mathcal{D}_{\text{src}}\cup\mathcal{D}_{\text{tar}}}\left[\exp(\alpha A(s, a))\log \pi_\eta(a|s)\right]$

29: **end for**

---

---

**Algorithm 2** DVDF-OTDF

---

1: **Input:** Source domain dataset $\mathcal{D}_{\text{src}}$, target domain dataset $\mathcal{D}_{\text{tar}}$, batch size $N$, data selection ratio $\xi$, alignment tradeoff coefficient $\lambda$
2: Initialize policy network $\pi_\phi$, value networks $V_\psi$, $Q_\theta$, target $Q$ function $Q_{\theta'}$, the cost function $C$, policy coefficients $\beta$, number of sampled latent variables $M$, target update rate $\eta$
3: **// Pre-train the advantage function**
4: Pre-train an SQL agent on $\mathcal{D}_{\text{src}}$, obtain $\hat{A}_{\text{pre}}(\cdot)$ and normalize $\hat{A}_{\text{pre}}(\cdot)$
5: **// Solve the OT problem**
6: Compute the optimal alignment between $\mathcal{D}_{\text{src}}$ and $\mathcal{D}_{\text{tar}}$ with Equation 4
7: Compute deviations $\{d_v\}_{v=1}^{|\mathcal{D}_{\text{src}}|}$ between the source domain data and $\mathcal{D}_{\text{tar}}$ with Equation 5
8: Normalize the deviations $d_v$ to obtain normalized deviations $d'_v$
9: Compute the score function $\omega(s, a, s') = \lambda \cdot d'_v(s, a, s') + (1 - \lambda) \cdot \hat{A}_{\text{pre}}(s, a, s')$
10: Concatenate $\mathcal{D}_{\text{src}}$ and $\{\omega_v\}_{v=1}^{|\mathcal{D}_{\text{src}}|}$ to get $\mathcal{D}'_{\text{src}} = \{(s_v, a_v, r_v, s'_v, \omega_v)\}_{v=1}^{|\mathcal{D}_{\text{src}}|}$
11: **for** $i = 1, 2, \dots$ **do**
12:     Sample a mini-batch $b_{\text{src}} := \{(s, a, r, s', w)\}_{v=1}^{N/2}$ from $\mathcal{D}'_{\text{src}}$
13:     Sample a mini-batch $b_{\text{tar}} := \{(s, a, r, s')\}_{v=1}^{N/2}$ from $\mathcal{D}_{\text{tar}}$
14:     Update the state value function $V_\psi$ via:
15:        $\mathcal{L}_V = \mathbb{E}_{(s,a) \sim \mathcal{D}_{\text{src}} \cup \mathcal{D}_{\text{tar}}} [L_2^\tau (Q_\theta(s, a) - V_\psi(s))]$
16:     **// Data filtering**
17:     Rank the deviations of the sample source domain data and reject the lowest $\xi\%$ of them
18:     Compute the weights for the remaining source domain data by $\exp(\beta \omega'_v)$
19:     Compute the target value via: $y = r + \gamma V_\psi(s')$
20:     Optimize the state-action value function $Q_\theta$ on $b_{\text{src}} \cup b_{\text{tar}}$ via:
21:        $\mathcal{L}_Q = \mathbb{E}_{(s,a,r,s') \sim \mathcal{D}_{\text{tar}}} \left[ (Q_\theta(s, a) - y)^2 \right] + \mathbb{E}_{(s,a,d) \sim \mathcal{D}'_{\text{src}}} \left[ \exp(\beta \omega'_v) (Q_\theta(s, a) - y)^2 \right]$
22:     Update the target network via: $\theta' \leftarrow \eta \theta + (1 - \eta) \theta'$
23:     **// Dataset regularization**
24:     Compute the advantage $A$ and optimize the policy $\pi_\phi$ on $b_{\text{src}} \cup b_{\text{tar}}$ using advantage-weighted regression (AWR) and dataset regulation:
25:        $\mathcal{L}_\pi = \mathbb{E}_{(s,a) \sim \mathcal{D}_{\text{src}} \cup \mathcal{D}_{\text{tar}}} [\exp(\beta A) \log \pi_\phi(a|s) - \beta \cdot \text{KL} (\pi_\phi(\cdot|s) \| \pi_{\text{prior}}(\cdot|s))]$
26: **end for**

---

Table 2: Hyperparameter setup for DVDF and baseline methods

| Hyperparameter | Value |
|---|---|
| **Shared** | |
| Actor network | (256, 256) |
| Critic network | (256, 256) |
| Learning rate | $3 \times 10^{-4}$ |
| Optimizer | Adam (Kingma & Ba, 2014) |
| Discount factor | 0.99 |
| Nonlinearity | ReLU |
| Target update rate | $5 \times 10^{-3}$ |
| Source domain Batch size | 128 |
| Target domain Batch size | 128 |
| **IQL** | |
| Temperature coefficient | 0.2 |
| Maximum log std | 2 |
| Minimum log std | -20 |
| Inverse temperature parameter $\beta$ | 3.0 |
| Expectile parameter $\tau$ | 0.7 |
| **DARA** | |
| Temperature coefficient | 0.2 |
| Classifier network | (256, 256) |
| Reward penalty coefficient $\lambda$ | 0.1 |
| **BOSA** | |
| Temperature coefficient | 0.2 |
| Maximum log std | 2 |
| Minimum log std | -20 |
| Policy regularization coefficient $\lambda_{\text{policy}}$ | 0.1 |
| Transition coefficient $\lambda_{\text{transition}}$ | 0.1 |
| Threshold parameter $\rho$ | $\log(0.01)$ |
| Value weight $\sigma$ | 0.1 |
| CVAE ensemble size of the dynamics model | 5 |
| **IGDF** | |
| Representation dimension | $\{16, 64\}$ |
| Contrastive encoder network | (256, 256) |
| Encoder pretrained steps | 7000 |
| Importance coefficient | 1.0 |
| Data selection ratio $\xi$ | 25% |
| **OTDF** | |
| CVAE training steps | 10000 |
| Number of sampled latent variables $M$ | 10 |
| Cost function | cosine |
| Data filtering ratio $\xi$ | 20% |
| **DVDF-IGDF** | |
| SQL pre-training steps | $1 \times 10^{6}$ |
| Trade-off coefficient $\lambda$ | 0.7 |
| Data selection ratio $\xi$ | 50% |
| **DVDF-OTDF** | |
| SQL pre-training steps | $1 \times 10^{6}$ |
| Trade-off coefficient $\lambda$ | 0.7 |
| Data filtering ratio $\xi$ | 50% |

Table 3: **Performance comparison under morphology shifts.** half=halfcheetah, hopp=hopper, walk=walker2d, r=random, m=medium, me=medium-expert, mr=medium-replay, e=expert. We report the normalized score evaluated in the target domain and $\pm$ captures the standard deviation across 5 seeds. We **bold** the highest scores for each task.

| Dataset | IQL | BOSA | DARA | IGDF | DVDF-IGDF | OTDF | DVDF-OTDF |
|---|---|---|---|---|---|---|---|
| half-r | **6.7** | 2.2 | 2.9 | **4.9±0.3** | 4.8±0.1 | **2.2±0.2** | 2.0±0.1 |
| half-m | 45.8 | 41.3 | 45.6 | 45.5±0.1 | **46.0±0.3** | **44.3±0.2** | 42.5±0.2 |
| half-mr | 26.1 | 27.8 | 28.9 | 24.2±3.3 | **31.1±4.7** | 19.7±2.5 | **27.2±1.3** |
| half-me | 63.0 | 44.4 | 59.2 | 50.2± 3.4 | **61.9±4.9** | 42.9±3.6 | **53.8±4.9** |
| half-e | 65.2 | 78.6 | 55.4 | 43.0±6.2 | **51.7±6.8** | 74.2±5.0 | **91.7±7.0** |
| hopp-r | 4.7 | 1.4 | 4.8 | **4.8±0.2** | 4.7±0.1 | **2.4±0.1** | 1.4±0.1 |
| hopp-m | 56.4 | 28.7 | 49.5 | **55.5±2.9** | 52.7±4.6 | 49.1±2.2 | **59.4±3.7** |
| hopp-mr | 51.3 | 40.6 | 53.5 | 54.9±5.8 | **58.6±6.4** | 24.9±3.4 | **32.6±4.5** |
| hopp-me | 35.8 | 20.2 | 38.2 | 43.3±3.6 | **61.2±4.2** | 51.8±3.9 | **63.4±5.3** |
| hopp-e | 87.2 | 64.3 | 77.1 | 51.5±2.9 | **86.9±4.2** | **113.2±5.9** | 109.5±2.1 |
| walk-r | 2.0 | 1.9 | 3.9 | 2.2±0.1 | **4.6±0.7** | 0.0±0.0 | 0.0±0.0 |
| walk-m | 32.6 | 40.3 | 25.0 | 33.0±2.3 | **62.3±6.1** | 40.3±7.1 | **61.7±9.2** |
| walk-mr | 9.0 | 2.9 | 6.9 | 9.5±0.4 | **13.6±1.2** | 14.1±1.8 | **18.8±1.6** |
| walk-me | 27.6 | 46.7 | 42.2 | 75.7±11.8 | **95.3±4.6** | 66.7±5.3 | **73.4±6.7** |
| walk-e | 103.4 | 30.2 | 102.7 | **108.3±6.7** | 103.5±5.9 | 103.5±1.9 | **108.8±3.2** |
| ant-r | 13.6 | **31.3** | 26.8 | 14.4±1.6 | **16.0±1.7** | 12.4±2.2 | **21.6±2.0** |
| ant-m | 89.1 | 36.1 | 96.4 | 91.6±4.4 | **101.1±5.9** | 92.5±2.7 | **102.7±3.4** |
| ant-mr | 59.7 | 24.0 | 64.1 | 58.2±7.1 | **64.8±4.6** | 69.6±8.1 | 57.4±2.0 |
| ant-me | 113.1 | 100.5 | 111.9 | 116.8±3.5 | **121.2±3.8** | 107.3±4.4 | **120.5±2.9** |
| ant-e | 116.3 | 76.3 | 124.5 | 126.8±1.7 | **129.0±2.4** | **111.0±2.4** | 107.9±4.0 |
| **Total** | 1008.6 | 739.7 | 1019.5 | 1039.0 | **1198.7** | 1042.1 | **1156.3** |

are available. This setting reflects real-world scenarios, such as nuclear power plant control, where accessing more target domain data is often impractical. Typical offline RL will fail under such extreme data scarcity, making the proper utilization of source domain data much more crucial.

**Tasks and Datasets.** The tasks and types of dynamics shifts are identical to those in Section 6.1. The only distinction lies in the target domain datasets, which now consist of only 5,000 transitions sampled from the D4RL datasets, instead of the 10% subset used in Section 6.1.

**Baselines.** We maintain the same baselines (IQL, BOSA, DARA, IGDF, and OTDF), and implement DVDF-IGDF and DVDF-OTDF for comparison as in Section 6.1.

**Experimental Results.** We run each algorithm for 1M gradient steps with 5 random seeds. We present the empirical results under kinematic shifts in Table 5, and the results under morphology shifts in Table 4.

As shown in Table 5, DVDF substantially enhances the performance of base algorithms, elevating total normalized scores by **26.2%** (IGDF) and **27.1%** (OTDF) under kinematic shifts. Specifically, DVDF-IGDF outperforms IGDF in **13** out of 20 tasks, and DVDF-OTDF surpasses OTDF in **12** out of 20 tasks, while achieving comparable performance in the remaining tasks.

The results in Table 4 demonstrate that DVDF maintains superiority over baseline methods under morphology shifts: DVDF-OTDF achieves the highest total score of **373.5** among all methods, surpassing OTDF by **12.6%** and leading in **13** out of 20 tasks. Similarly, DVDF-IGDF improves IGDF by **15.0%** in total score and achieves better performance in **12** out of 20 tasks. These results demonstrate the superiority of DVDF with extremely limited target domain data.

### E.3 MORE COMPARISONS WITH RECENT STUDIES

In this section, we compare our method DVDF with two more recent studies, PSEC (Liu et al., 2025) and DmC (Le Pham Van et al., 2025). PSEC proposes to preserve the prior learned skills in a parametric space and adaptively composes them using a context-aware module to handle new tasks.

Table 4: **Performance comparison under morphology shifts with extremely limited target domain data.** We report the normalized score evaluated in the target domain, and $\pm$ captures the standard deviation across 5 seeds. We **bold** the highest scores for each task.

| Dataset | IQL | BOSA | DARA | IGDF | DVDF-IGDF | OTDF | DVDF-OTDF |
|---|---|---|---|---|---|---|---|
| half-r | 0.0 | **2.2** | 2.0 | 0.0±0.0 | 0.0±0.0 | 2.0±0.1 | **2.2±0.1** |
| half-m | 18.7 | 17.3 | 16.1 | 22.6±1.2 | **26.7±3.5** | **24.6±3.4** | 22.9±3.6 |
| half-mr | 12.5 | 9.5 | 8.6 | 14.8±1.9 | **19.4±2.0** | 17.9±1.6 | **25.1±2.4** |
| half-me | 12.3 | 15.4 | 13.7 | 14.9±0.5 | **21.9±3.1** | 11.5±0.8 | **19.8±1.7** |
| half-e | 4.9 | 3.6 | 2.9 | **6.2±0.1** | 5.9±0.2 | 10.7±3.5 | **15.4±2.2** |
| hopp-r | 3.7 | 1.1 | 3.4 | **4.1±0.4** | 3.8±0.1 | **4.4±0.2** | 4.0±0.1 |
| hopp-m | **35.2** | 20.6 | 25.5 | 31.6±4.2 | 20.3±2.9 | 24.2±3.8 | 19.3±2.0 |
| hopp-mr | 2.3 | 3.7 | 3.5 | 4.1±0.3 | **7.4±0.4** | 4.6±0.2 | **5.6±0.3** |
| hopp-me | **38.3** | 10.2 | 19.7 | 36.3±3.7 | **43.2±2.8** | 31.6±2.9 | 37.4±3.8 |
| hopp-e | 28.3 | 7.3 | 13.0 | 29.6±2.0 | **44.6±7.6** | 43.3±6.2 | **48.9±4.1** |
| walk-r | 0.0 | 0.0 | 0.0 | 0.0±0.0 | 0.0±0.0 | 0.0±0.0 | 0.0±0.0 |
| walk-m | 16.4 | 10.6 | 15.8 | 14.3±2.3 | **24.3±1.2** | 19.3±2.4 | **23.7±2.2** |
| walk-mr | 3.6 | 0.0 | 2.9 | **4.4±0.6** | 3.0±0.2 | 4.1±0.5 | **4.8±0.6** |
| walk-me | 16.7 | 12.8 | 10.2 | 12.6±1.0 | **20.9±3.7** | 15.4±1.2 | **23.0±1.5** |
| walk-e | 8.3 | 9.3 | 12.4 | **13.9±0.1** | 10.2±0.4 | 13.5±0.4 | **18.9±0.2** |
| ant-r | 14.1 | **20.3** | 16.2 | 13.2±0.4 | **17.1±3.6** | **10.2±0.6** | 9.1±0.2 |
| ant-m | 17.3 | 30.1 | **32.9** | 25.6±2.5 | 28.1±5.5 | **32.3±4.0** | 26.4±4.4 |
| ant-mr | **29.8** | 19.7 | 13.5 | 28.7±1.5 | 19.7±1.9 | 20.4±3.0 | **27.0±2.2** |
| ant-me | 15.4 | 15.8 | 12.3 | 17.5±2.2 | **21.1±2.8** | 19.1±1.8 | **23.2±1.9** |
| ant-e | 20.7 | 20.5 | **23.1** | 15.8±1.1 | **19.0±3.2** | **22.7±1.4** | 16.8±0.8 |
| **Total** | 298.5 | 230.0 | 247.7 | 310.2 | **356.6** | 331.8 | **373.5** |

In the cross-domain offline setting, PSEC first learns separate policies from the source domain and target domain data, which are then dynamically combined to work under the target dynamics. DmC employs a KNN-based estimator as a measure of dynamics gap, and utilizes the KNN proximity score as a guiding signal for diffusion-based data generation. Source domain samples are selected based on the proximity score and combined with the target data for training. As a plug-in method, DVDF could be seamlessly integrated into both PSEC and DmC. Specifically, DVDF could assist PSEC by selecting beneficial source domain samples to facilitate target policy learning from the limited target dataset. Similarly, DVDF could be integrated into DmC's source data selection process to enable dynamics- and value-aligned data selection. We refer to these two integrated methods as DVDF-PSEC and DVDF-DmC.

We evaluate DVDF-PSEC and DVDF-DmC on four tasks (`halfcheetah`, `hopper`, `walker2d`, `ant`) under kinematic shifts, with datasets of three qualities: `medium`, `medium-expert`, and `expert`. All other experimental settings follow Section 6.1. The performance comparison between the base algorithms (PSEC, DmC) and their DVDF-enhanced versions (DVDF-PSEC and DVDF-DmC) is presented in Table 6.

The results in Table 6 show that DVDF-PSEC outperforms PSEC on 11 out of 12 datasets, while DVDF-DmC surpasses DmC on 10 out of 12. Furthermore, the DVDF-enhanced versions achieve a substantially higher total score, confirming the versatility of DVDF as a plug-in module.

# F  COMPUTE INFRASTRUCTURE

We list our compute infrastructure for our experiments in Table 7.

Table 7: Compute Infrastructure

| CPU | GPU | Memory |
|---|---|---|
| AMD EPYC 7452 | RTX3090×8 | 288GB |

Table 5: **Performance comparison under kinematic shifts with extremely limited target domain data.** We report the normalized score evaluated in the target domain and $\pm$ captures the standard deviation across 5 seeds. We **bold** the highest scores for each task.

| Dataset | IQL | BOSA | DARA | IGDF | DVDF-IGDF | OTDF | DVDF-OTDF |
|---|---|---|---|---|---|---|---|
| half-r | 4.8 | 2.2 | **6.7** | 5.6±1.4 | 4.8±0.5 | **2.1±0.1** | 1.7±0.1 |
| half-m | 19.8 | 23.6 | 20.4 | 21.6±0.7 | **29.7±1.6** | **22.8±1.9** | 21.3±2.6 |
| half-mr | 5.9 | 0.0 | 4.0 | **7.7±1.2** | 6.6±2.1 | 4.0±1.1 | **9.3±1.5** |
| half-me | 9.5 | 11.1 | 7.2 | 14.3±0.9 | **22.9±1.0** | 7.6±0.4 | **13.9±3.7** |
| half-e | **7.3** | 4.2 | 6.1 | 4.2±0.1 | **6.1±0.1** | **5.2±1.6** | 4.7±1.0 |
| hopp-r | 2.4 | 1.9 | 2.2 | 3.7±0.2 | **4.2±0.1** | 1.2±0.1 | **5.1±1.4** |
| hopp-m | 26.1 | 10.6 | 13.2 | 34.6±5.9 | **38.4±4.1** | 36.1±4.4 | 32.8±3.1 |
| hopp-mr | 7.4 | 7.8 | 9.8 | 9.8±1.0 | **13.0±2.9** | 14.7±3.3 | **21.3±5.0** |
| hopp-me | 9.3 | 11.4 | 8.6 | 12.3±1.4 | **20.1±5.2** | 7.1±2.1 | **15.2±2.9** |
| hopp-e | **11.1** | 8.3 | **11.8** | 9.4±0.5 | 8.0±0.2 | **6.7±0.3** | 6.4±0.1 |
| walk-r | 4.6 | 0.0 | 0.0 | **8.1±2.9** | 6.6±1.4 | 0.0±0.0 | 0.0±0.0 |
| walk-m | 7.7 | 7.6 | 4.4 | 14.3±1.7 | **23.0±3.9** | 11.8±1.9 | **16.1±3.7** |
| walk-mr | 3.9 | 9.1 | 4.3 | 2.4±0.1 | **3.7±0.1** | 7.4±1.3 | **16.0±1.6** |
| walk-me | 5.7 | 4.8 | 6.4 | 8.4±2.1 | **16.2±4.4** | 8.1±2.4 | **15.9±3.8** |
| walk-e | 10.6 | 9.3 | 20.1 | **13.7±2.8** | 11.9±1.6 | 15.8±2.0 | **19.3±1.2** |
| ant-r | 7.0 | 6.5 | 5.5 | **11.8±3.0** | 8.4±2.2 | 7.3±0.5 | **10.1±0.6** |
| ant-m | 14.6 | 19.1 | 21.3 | 20.3±1.2 | **24.1±3.3** | 42.3±7.7 | **48.1±6.3** |
| ant-mr | 7.3 | **17.6** | 13.2 | 3.9±0.7 | **13.4±2.6** | 17.6±2.8 | 14.7±2.3 |
| ant-me | 5.3 | 10.1 | 2.9 | **9.4±4.6** | 9.0±4.8 | 4.3±0.5 | **12.3±4.6** |
| ant-e | 3.1 | 4.3 | 0.0 | 2.9±1.4 | **5.5±0.8** | **5.1±1.2** | 4.7±1.5 |
| **Total** | 173.4 | 169.5 | 141.3 | 218.4 | **275.6** | 227.2 | **288.9** |

Table 6: **Performance comparison with PSEC and DmC under kinematic shifts.** We report the normalized score evaluated in the target domain and $\pm$ captures the standard deviation across 5 seeds. We **bold** the highest scores for each task.

| Dataset | PSEC | DVDF-PSEC | DmC | DVDF-DmC |
|---|---|---|---|---|
| half-kine-m | **33.4±1.0** | 30.1±0.8 | **39.6±1.1** | 33.2±0.2 |
| half-kine-me | 41.6±2.3 | **49.3±2.7** | 47.3±5.8 | **52.6±3.2** |
| half-kine-e | 52.0±4.7 | **58.3±3.4** | 66.9±2.4 | **73.6±1.6** |
| hopper-kine-m | 47.2±3.5 | **53.2±1.5** | 53.8±5.0 | **65.4±2.8** |
| hopper-kine-me | 31.7±3.6 | **42.0±2.4** | 47.2±1.3 | **56.3±2.8** |
| hopper-kine-e | 70.1±4.3 | **74.2±3.4** | 92.6±2.7 | **98.2±1.6** |
| walker2d-kine-m | 41.0±1.9 | **56.7±5.2** | 48.3±4.8 | **65.1±2.4** |
| walker2d-kine-me | 53.7±3.0 | **58.6±3.6** | 62.6±2.9 | **69.2±1.2** |
| walker2d-kine-e | 75.9±1.4 | **95.8±2.8** | 83.0±2.6 | **93.0±2.6** |
| ant-kine-m | 82.2±1.5 | **87.4±1.4** | **84.1±5.5** | 81.2±2.0 |
| ant-kine-me | 94.6±1.7 | **106.0±2.9** | 107.1±3.5 | **112.3±2.1** |
| ant-kine-e | 103.5±1.0 | **118.4±1.5** | 101.2±1.0 | **116.1±2.9** |
| **Total** | 726.9 | **830.0** | 833.7 | **916.2** |

## G  TRAINING TIME

We report the average training time for our method and the baselines (including IQL, IGDF, OTDF, DVDF-IGDF and DVDF-OTDF) across various tasks over 1M steps in Table 8. Note that the additional computational overhead for OTDF and DVDF stems from solving complex optimal transport matrices and pre-training for the advantage function, respectively. Fortunately, these computations can be precomputed, minimizing their impact on subsequent experiments.

Table 8: Training time comparison between various methods. h=hour(s), m=minute(s).

| IQL | IGDF | OTDF | DVDF-IGDF | DVDF-OTDF |
|-----|------|------|-----------|-----------|
| 5h24m | 6h56m | 9h17m | 11h43m | 14h07m |

## H    BROADER IMPACTS

This paper presents work whose goal is to promote effective cross-domain offline RL. Our work has potential positive social impacts. For example, our research could enable more efficient development of advanced robotics systems by effectively utilizing source domain data. At present, we have not identified any foreseeable negative impacts arising from this research.

## I    LLM USAGE DECLARATION

The use of LLMs in this work is strictly limited to grammatical polishing of the initial draft. LLMs are not involved in any core research components, including but not limited to the conception of the method, theoretical proofs, and experiments.

