# OpenReview forum: "Cross-domain Offline Policy Adaptation with Dynamics- and Value-aligned Data Filtering"
_ICLR.cc/2026/Conference — Submitted to ICLR 2026_

### Official Review · Reviewer_hTDh · 2025-10-29

**Soundness:** 2
**Presentation:** 3
**Contribution:** 2
**Rating:** 4
**Confidence:** 4

**Summary:**

Cross-domain offline RL aim to train an agent for the target domain using data from both source and target domains. Previous methods selectively sharing source domain samples that exhibit dynamic alignment with the target domain. This paper argues that focusing solely on dynamics alignment overlooks value alignment, the selection of high-quality, high-value samples from the source domain. It proposes a method to selectively utilize these source domain samples with both high dynamic and value alignment, using a combination of a dynamic score and an advantage function. This approach results in the Dynamics- and Value-aligned Data Filtering (DVDF) method. Theoretical analysis and experiments are conducted to validate the effectiveness of this method.

**Strengths:**

- The paper is clearly motivated and well-written, with a logical structure that guides readers through the problem setup and proposed solution.

- The proposed method is reasonable, experiments demonstrate the effectiveness of the method.

**Weaknesses:**

- The practical implementation introduces several approximations to the analysis, which prevents the value misalignment term from being controlled in practice. Additionally, the introduction of $\hat{A}$ and SQL is not very clear to me. For instance, it is unclear why an additional advantage function is needed, and why SQL outperforms IQL.

- In principle, a more accurate advantage is better. While the advantages of SQL and IQL shown in Figure 2(a) indicate that SQL’s advantage is higher than IQL’s, this does not mean SQL achieves a more accurate advantage. Furthermore, Figure 2(b) exhibits high variance with only 5 seeds, making the results unconvincing.

- The score function introduces a parameter lambda that trades off value alignment and dynamic alignment. However, the experiments in Figure 3 do not reflect the importance of value alignment. There are no significant differences in the results (ranging from ~90 to ~110, and ~60 to ~70), and high variance with only 5 seeds further obscures these minor differences.

- The ablation study lacks some important components that would help better understand the method. For example, Figure 2 omits Solely dynamics, and Figure 3 omits the cases of lambda=1 and lambda=0. Moreover, in my view, Figure 2(b) and Figure 3(a) could be combined into a single figure.

**Questions:**

- The proposed method includes two terms for selecting source-domain data. I am curious how differences in dynamics affect the importance of these two terms. Furthermore, is there a relationship between parameter selection and dynamics differences?

- As a plug-in module, integrating DVDF into IGDF and OTDF leads to performance degradation on certain tasks—such as half-r and ant-mr. What is the reason for this?

- The value misalignment term only relates to the source domain. Does this mean the expert dataset will have more samples selected than the random and medium datasets? Moreover, if the dataset is expert-level, does this imply that reinforcement learning algorithms (whether IQL or SQL) are unnecessary, and supervised learning alone is sufficient?

- In my understanding, VGDF is an implicit combination of value alignment and dynamic alignment. How should we interpret the paper’s claim that "VGDF still only addresses dynamics mismatch"?

- Why do SQL provide more reliable advantage estimates than IQL? Additionally, how do other offline RL methods such as extreme Q-learning and in-sample actor-critic perform in this regard?

- See Weaknesses.

---

> ### Author Response · Authors · 2025-11-21
> **Reply to Reviewer hTDh (Part 1)**
>
> We thank the reviewer for the thoughtful review and for acknowledging the strengths of our work. We present our point-to-point responses to the major concern below. We hope our responses could address the reviewer's concerns.
>
> ## Concern 1: The practical implementation introduces approximations to the analysis, and the introduction of advantage function and SQL is not clear
>
> We first address the concern on why $\hat{A}$ and SQL are necessary. The advantage function $\hat{A}$ is for the measure of value misalignment term. As we derive in Proposition 5.1, the value misalignment could be lower bounded by the in-sample optimal advantage value on the source domain data. Therefore, we need to pretrain an advantage function $\hat{A}$ to approximate the in-sample optimal advantage function to measure the value misalignment term for data filtering. Intuitively, we prefer source domain samples with a higher advantage value to minimize the value misalignment. SQL and IQL are algorithms for pre-training to obtain the advantage function. We choose SQL instead of IQL since IQL exhibits severe V-function underestimation, and the learned advantage function is not accurate. **This is evidenced in Page 8 in SQL paper [1]:** "In IQL, the derivative keeps decreasing as the residual becomes more negative, hence, the V-function will be over-underestimated by those bad actions whose Q-value is extremely small". SQL addresses this issue by assigning a zero probability mass to those bad actions, enabling a more accurate advantage estimation.
>
> We then address the concern on approximations to the analysis. The major approximation we make in our implementation is to use a pre-trained advantage function to approximate the ideal in-sample optimal advantage function. However, we analyse the advantage approximation error in Proposition 5.2, and propose to use SQL for pre-training to minimize the approximation error. Thus, we believe our implementation is well-motivated and remains closely aligned with our theoretical analysis.
>
> ## Concern 2: Figure 2(a) does not show SQL achieves a more accurate advantage than IQL
>
> We appreciate this suggestion. Superior performance does not necessarily indicate that SQL learns a more accurate advantage function than IQL. To address this concern, we empirically compare the advantage functions learned by SQL and IQL based on the advantage estimation error recorded during pre-training.  The advantage estimation error is defined as: $\\mathcal{E}=\\mathbb{E} _ {(s,a)\\in\\mathcal{D} _ {src}}\\frac{\\hat{A}(s,a)-A(s,a)}{A(s,a)}$, where $\hat{A}$ is the estimated advantage by SQL or IQL, and $A$ is the true advantage computed by Monte Carlo rollouts. a positive $\mathcal{E}$ indicates that the advantage is overestimated, and a negative $\mathcal{E}$ means the advantage is underestimated.
>
>
> We conduct experiments on four datasets for evaluation: halfcheetah-morph-expert, halfcheetah-morph-medium-expert, halfcheetah-kinematic-medium and halfcheetah-kinematic-medium-replay, and we have provided the recorded advantage estimation error curves in Section 6.2 in our revision. We also present the advantage estimation error at the last step of pre-training for SQL and IQL below.
>
> ||half-morph-e|half-morph-me|half-kinematic-m|half-kinematic-mr|
> |-|-|-|-|-|
> |IQL|1.21$\pm$0.01|2.07$\pm$0.02|2.37$\pm$0.01|2.12$\pm$0.57|
> |SQL|-0.83$\pm$0.01|-0.86$\pm$0.01|-0.71$\pm$0.01|-0.49$\pm$0.01|
>
> We find that IQL tends to overestimate the advantage value, consistent with SQL paper [1], which notes that IQL tends to underestimate the $V$-function, causing advantage function overestimation. In contrast, SQL typically underestimates the advantage since it incorporates sparsity in $V$-function learning. Crucially, the absolute estimation error of SQL is consistently smaller than that of IQL, demonstrating that SQL provides a more accurate advantage function for our method.

---

> ### Author Response · Authors · 2025-11-21
> **Reply to Reviewer hTDh (Part 2)**
>
> ## Concern 3: No significant performance difference for different $\lambda$ in Figure 3
>
> We appreciate the reviewer's observation. The sensitivity of the hyperparameter $\lambda$ is dependent on the specific task and dataset. To further demonstrate this point, we conduct a sensitivity analysis of $\lambda$ on four datasets: hopper-kinematic-me, hopper-kinematic-e, ant-morph-m, and ant-morph-mr. We choose DVDF-IGDF as our base algorithm, and $\lambda$ is ranged across {0, 0.3, 0.5, 0.7, 0.9, 1.0}. To reduce the performance variance, we run the experiments with 10 random seeds. We report the final performance under different values of $\lambda$ on 4 datasets below:
>
> |Datasets|$\\lambda=0$|$\\lambda=0.3$|$\\lambda=0.5$|$\\lambda=0.7$|$\\lambda=0.9$|$\\lambda=1.0$|
> |-|-|-|-|-|-|-|
> |hopper-kinematic-me|22.9$\pm$5.2 |34.0$\pm$5.1|**63.4$\pm$4.8**|61.2$\pm$4.0|29.7$\pm$7.6|**14.6$\pm$4.5**|
> |hopper-kinematic-e|70.2$\pm$5.2 |**92.4$\pm$6.3** |85.0$\pm$3.0 |86.2$\pm$4.6| 72.6$\pm$5.5|**68.9$\pm$4.0**|
> |ant-morph-m|87.2$\pm$3.0|**83.1$\pm$4.5**|94.2$\pm$6.6|**103.3$\pm$5.1**|89.6$\pm$5.0|95.7$\pm$6.4|
> |ant-morph-mr|59.1$\pm$3.5|**54.2$\pm$3.4**|**68.2$\pm$4.6**|65.1$\pm$4.7|64.3$\pm$3.8|61.0$\pm$3.0|
>
> We highlight in bold the lowest and highest scores over different $\lambda$ values for each dataset. The results indicate that datasets exhibit varying sensitivities to $\lambda$. For example, performance on the ant-morph-mr dataset shows less variation across different $\lambda$ values (ranged from 54.2 to 68.2), whereas performance on the hopper-kinematic-me dataset is more sensitive to $\lambda$ (ranged from 14.6 to 63.4). We hypothesize that this difference arises because medium-expert datasets typically contain a higher proportion of high-quality samples than medium-replay datasets. As $\lambda$ controls the weights of these high-quality samples, its effect becomes more pronounced on datasets where such samples are abundant. We have included the above results in our parameter sensitivity part (Section 6.3) in our revision.
>
> ## Concern 4: The ablation study lacks some important components such as $\lambda=0$ and $\lambda=1$
>
> We have taken the reviewer's suggestion and incorporated the second part of the ablation study (on value-aligned data filtering) into the parameter sensitivity section in our revision. Filtering solely based on value alignment is equivalent to setting $\lambda=0$, and filtering solely based on dynamics alignment is equivalent to $\lambda=1$. We believe this concern could be addressed by our responses to Concern 3 and the analysis in Figure 3(a) of our revision.
>
> ## Concern 5: How differences in dynamics affect the importance of dynamics and value alignment, and is there a relationship between parameter selection and dynamics differences?
>
> We thank the reviewer for this insightful question. We believe the difference in dynamics would affect the importance of dynamics and value alignment. For instance, if a source dataset consists of expert-level and random-level samples and exhibits minor dynamics mismatch with the target domain, then value alignment is more significant, and dynamics alignment can be largely ignored. However, if there is a dynamics mismatch, then we have to consider dynamics alignment instead of solely considering value alignment. This could be verified by our extended results of the motivation example in Appendix D.2, where we find that DVDF outperforms Value-IGDF (which selects source domain samples solely based on value alignment).
>
> It is hard to establish an exact analytical relationship between the degree of the dynamics difference and the optimal parameter (i.e., $\lambda$). As a general guideline, a larger dynamics mismatch would necessitate a larger $\lambda$ to emphasize dynamics alignment. However, this is not an absolute rule, as the optimal $\lambda$ also depends on the quality of the source dataset. For instance, an expert-level dataset would benefit from a smaller $\lambda$ to prioritize its high-value samples.
> Consequently, the optimal $\lambda$ cannot be determined without some task-specific tuning. However, our experimental results indicate that a default value of $\lambda=0.7$ works generally well across most tasks we evaluate. Therefore, for a new task, we recommend starting with $\lambda=0.7$ and then conducting a finer-grained search if optimal performance is required.

---

> ### Author Response · Authors · 2025-11-21
> **Reply to Reviewer hTDh (Part 3)**
>
> ## Concern 6: DVDF leads to performance degradation on certain tasks
>
>
> We attribute the performance degradation of DVDF on certain tasks to the fact that different tasks have different optimal $\lambda$ values. Recall that the baseline IGDF/OTDF method is equivalent to setting $\lambda=1.0$. In our study, we use a fixed $\lambda=0.7$ for all tasks to reduce the tuning burden. It is hard to achieve the best performance on all the tasks with a fixed $\lambda$.  Since **the core contribution of this work is to reveal the role of value alignment in cross-domain offline RL instead of achieving a SOTA performance on all tasks**, we **do not extensively tune $\lambda$** and maintain a fixed value of 0.7 across all datasets, which could account for DVDF's performance degradation on certain datasets. We believe **a more sophisticated task-specific tuning would lead to better performance**.
>
> ## Concern 7: Will the expert dataset have more samples selected, and whether supervised learning is sufficient?
>
> **In our method, the absolute number of samples selected from any dataset is determined by the data selection ratio $\xi$, not by the dataset quality**. We understand the reviewer's question as concerning the relative probability of selection for samples of different quality levels within a mixed dataset (e.g., medium-expert). **This probability depends critically on $\lambda$**.
> When $\lambda=0$ (pure value alignment), expert samples have a higher selection probability than medium or random samples.
> When $\lambda>0$, medium or random samples with a small dynamics mismatch may be selected over an expert sample with a large mismatch.
>
> We argue that **supervised learning (i.e., behavior cloning) is insufficient for the cross-domain offline RL problem**. Even with an expert-level source dataset, behavior cloning cannot outperform the behavior policy (resulting in a normalized score less than 100). However, as our results in Table 1 show, RL-based methods can achieve scores above 100 (e.g., on ant-expert). This clearly demonstrates that RL is necessary to achieve a better performance.
>
>
> ## Concern 8: How to interpret that VGDF only addresses the dynamics mismatch
>
> We emphasize that the term "value alignment" has a fundamentally different meaning for VGDF and DVDF.
> **In VGDF, the value alignment means minimizing $|V(s^\prime_{src})-V(s^\prime_{tar})|$, the objective is to prevent significant value estimation errors during Bellman backups**. This process implicitly quantifies the dynamics mismatch from a value-function perspective and is unrelated to the inherent optimality or quality of the data samples themselves. Therefore, we characterize VGDF as a method that addresses the dynamics mismatch. **In contrast, value alignment in DVDF refers to tightening the performance difference between the learned policy and the in-sample optimal policy in the source domain**. **This principle directly motivates the selection of source data samples that are of higher quality (i.e., more optimal)**. This concept of sample optimality is orthogonal to dynamics mismatch as considered in VGDF.

---

> ### Author Response · Authors · 2025-11-21
> **Reply to Reviewer hTDh (Part 4)**
>
> ## Concern 9: Why does SQL provide more accurate advantage estimation than IQL? and how do other offline RL methods perform?
>
> SQL provides a more accurate advantage estimation than IQL, since IQL exhibits severe $V$-function underestimation, and the learned advantage function is not accurate. **This is evidenced in Page 8 in SQL paper [1]**: "In IQL, the derivative keeps decreasing as the residual becomes more negative, hence, the $V$-function will be over-underestimated by those bad actions whose Q-value is extremely small". SQL addresses this issue by assigning a zero probability mass to those bad actions, enabling a more accurate advantage estimation.
>
> For the second question, we evaluate the advantage estimation of IAC [2] and XQL [3]. The advantage estimation errors for these methods at the end of the pre-training phase are shown below, alongside the results for IQL and SQL from our response to Concern 2:
>
> ||half-morph-e|half-morph-me|half-kinematic-m|half-kinematic-mr|
> |-|-|-|-|-|
> |IQL|1.21$\pm$0.01|2.07$\pm$0.02|2.37$\pm$0.01|2.12$\pm$0.57|
> |SQL|-0.83$\pm$0.01|-0.86$\pm$0.01|-0.71$\pm$0.01|-0.49$\pm$0.01|
> |IAC|-0.92$\pm$0.01|-1.12$\pm$0.01|-0.77$\pm$0.01|-0.83$\pm$0.01|
> |XQL|-0.80$\pm$0.01|-0.87$\pm$0.01|-0.69$\pm$0.01|-0.52$\pm$0.01|
>
> The results indicate that IAC and XQL, like SQL, tend to underestimate the advantage value, but their estimation errors are consistently smaller than those of IQL. This improvement can be attributed to their algorithmic designs: IAC avoids $V$-function estimation entirely by learning only a $Q$-function, while XQL assigns an exponentially small probability mass to bad actions during $V$-function learning, thereby reducing their negative influence compared to IQL.
>
>
> [1] Offline RL with No OOD Actions: In-Sample Learning via Implicit Value Regularization. ICLR 2023
>
> [2] In-sample Actor Critic for Offline Reinforcement Learning. ICLR 2023
>
> [3] Extreme Q-Learning: MaxEnt RL without Entropy. ICLR 2023

---

> > ### Author Response · Authors · 2025-11-27
> >
> > Dear Reviewer hTDh, thanks for your thoughtful review. As the author-reviewer discussion period is near its end, we wonder if our rebuttal addresses your concerns. Please let us know if any further clarifications or discussions are needed!

---

### Official Review · Reviewer_qZcZ · 2025-10-30

**Soundness:** 3
**Presentation:** 2
**Contribution:** 3
**Rating:** 4
**Confidence:** 3

**Summary:**

This paper proposes a Dynamics- and Value-aligned Data Filtering (DVDF) method, which can selectively share those source domain samples with both high dynamics and value alignment.

**Strengths:**

* From a theoretical perspective, this paper reveals that the existing theoretical framework that tightens the performance discrepancy of a given policy between the source and target domains misaligns with the RL objective, and fails to guarantee learning a well-performing target policy.

* DVDF trades off the dynamics and value misalignment and selectively shares source domain samples to train the policy.

* DVDF can be generally treated as a plug-in module and seamlessly integrated with recent methods like IGDF and OTDF.

**Weaknesses:**

* The paper does not clearly explain how the advantage function is obtained.

* The experimental evaluation lacks results on lower-quality datasets.

**Questions:**

* It would be helpful to clarify whether the advantage function is affected by overestimation and how accurately it can be estimated in practice.

* Have the authors considered validating the proposed approach through real-world robot experiments?

* Could the authors include comparisons and a discussion with recent cross-domain offline RL studies (e.g., PSEC, DmC)? Incorporating these baselines and analysing the differences would strengthen the paper and better position the proposed method within the current literature.

Reference:

Liu, T., Li, J., Zheng, Y., Niu, H., Lan, Y., Xu, X., Zhan, X. Skill expansion and composition in parameter space. In International Conference on Learning Representations, 2025.

Van, L. L. P., Nguyen, M. H., Kieu, D., Le, H., Tran, H. T., & Gupta, S. DmC: Nearest Neighbor Guidance Diffusion Model for Offline Cross-domain Reinforcement Learning. arXiv preprint arXiv:2507.20499.

---

> ### Author Response · Authors · 2025-11-21
> **Reply to Reviewer qZcZ (Part 1)**
>
> We thank the reviewer for the insightful review and for acknowledging the strengths of our work. Our responses to the major concerns are presented below. We hope our responses could address the reviewer's concerns.
>
> ## Concern 1: The paper does not explain how the advantage function is obtained
>
> The advantage function is computed directly by subtracting the state-value function $V(s)$ from the action-value function $Q(s,a)$. Specifically, SQL learns a $V(s)$ and a $Q(s,a)$ during training. We then obtain the advantage function by definition: $A(s,a)=Q(s,a)−V(s)$. We have clarified this point in our revision.
>
> ## Concern 2: The experimental evaluation lacks results on lower-quality datasets
>
> We thank the reviewer for this comment. We would like to clarify that **in Section 6.1 and Appendix E.1, we have already evaluated DVDF on datasets of varying quality, including random-quality data and an extremely low-data setting** (5,000 transitions in the target domain). These results demonstrate DVDF's effectiveness on low-quality datasets.
>
> To further strengthen this validation under more challenging conditions, we have conducted additional experiments with increased dynamics shifts. Specifically, we intensify the kinematic shifts in four tasks (halfcheetah, hopper, walker2d, ant) by modifying the .xml files as shown below.
>
>
> ```xml
> # halfcheetah
> # broken back thigh joint
> <joint axis="0 1 0" damping="6" name="bthigh" pos="0 0 0" range="-.00052 .00105" stiffness="240" type="hinge"/>
> ```
>
> ```xml
> # hopper
> # broken head joint
> <joint axis="0 -1 0" name="thigh_joint" pos="0 0 1.05" range="-0.015 0" type="hinge"/>
> # broken foot joint
> <joint axis="0 -1 0" name="foot_joint" pos="0 0 0.1" range="-9 9" type="hinge"/>
> ```
>
> ```xml
> # walker2d
> # broken right foot joint
> <joint axis="0 -1 0" name="foot_joint" pos="0 0 0.1" range="-0.045 0.045" type="hinge"/>
> ```
>
> ```xml
> # ant
> # broken hip joints of front legs
> <joint axis="0 0 1" name="hip_1" pos="0.0 0.0 0.0" range="-0.3 0.3" type="hinge"/>
> <joint axis="0 0 1" name="hip_2" pos="0.0 0.0 0.0" range="-0.03 0.03
> " type="hinge"/>
> ```
>
> The target domain datasets consist of 10\% samples from D4RL datasets, and we collect source domain datasets of three data qualities (medium, medium-expert, expert) following the same procedure as in Section 6.1. We apply DVDF to IGDF and OTDF for comparison. The experimental results are presented below.
>
> |Datasets|IGDF|DVDF-IGDF|OTDF|DVDF-OTDF|
> |-|-|-|-|-|
> |half-kine-m|**43.1$\pm$0.5**|41.2$\pm$0.1|43.0$\pm$0.1|**45.4$\pm$0.1**|
> |half-kine-me|51.8$\pm$4.6|**62.4$\pm$5.7**|**43.6$\pm$3.8**|41.5$\pm$2.9|
> |half-kine-e|45.7$\pm$2.4|**57.8$\pm$2.7**|70.6$\pm$3.0|**85.3$\pm$4.7**|
> |hopper-kine-m| 48.4$\pm$5.3|**56.5$\pm$2.6**|43.8$\pm$3.4|**61.5$\pm$4.5**|
> |hopper-kine-me|3.5$\pm$0.2|**53.6$\pm$5.8**|43.7$\pm$4.3|**62.3$\pm$6.6**|
> |hopper-kine-e|63.5$\pm$5.7|**80.6$\pm$4.4**|88.5$\pm$1.3|**102.4$\pm$4.6**|
> |walker2d-kine-m|50.4$\pm$2.1|**67.5$\pm$5.3**|34.9$\pm$1.4|**66.8$\pm$4.3**|
> |walker2d-kine-me|73.8$\pm$2.6|**88.5$\pm$4.2**|56.3$\pm$3.8|**79.4$\pm$2.1**|
> |walker2d-kine-e|90.9$\pm$1.1|**104.5$\pm$2.0**|86.7$\pm$3.4|**97.3$\pm$2.0**|
> |ant-kine-m|80.5$\pm$2.4|**92.3$\pm$3.9**|72.3$\pm$4.6|**85.2$\pm$5.7**|
> |ant-kine-me|113.5$\pm$2.3|**122.0$\pm$1.6**|100.4$\pm$1.3|**117.8$\pm$2.5**|
> |ant-kine-e|106.8$\pm$1.6|**111.7$\pm$2.2**|98.3$\pm$2.1|**105.5$\pm$3.4**|
> |total|771.9|**938.6**|782.1|**950.4**|
>
> The results show that DVDF consistently outperforms the base algorithms (IGDF, OTDF) across most datasets, achieving **significantly higher** total scores (938.6 vs. 771.9 for IGDF, and 950.4 vs. 782.1 for OTDF). These additional experiments provide further evidence of DVDF's robustness on lower-quality datasets under dynamics shifts.

---

> ### Author Response · Authors · 2025-11-21
> **Reply to Reviewer qZcZ (Part 2)**
>
> ## Concern 3: Whether the advantage function is affected by overestimation and how accurately it can be estimated
>
> Since IQL and SQL are in-sample offline RL algorithms without requiring out-of-distribution (OOD) actions, **no significant $V$-function _overestimation_ would occur** and affect the advantage estimation. However, as pointed out in page 8 in SQL paper [1]: "In IQL, the derivative keeps decreasing as the residual becomes more negative, hence, the $V$-function will be over-underestimated by those bad actions whose Q-value is extremely small", we conclude that **$V$-function _underestimation_ would affect the advantage estimation**. That is the reason we leverage SQL for pre-training, since SQL assigns a zero probability mass to those bad actions and enables a more accurate advantage estimation.
>
> We also empirically evaluate the advantage function learned with SQL and IQL, by recording the advantage estimation error during pre-training. The advantage estimation error is computed as: $\\mathcal{E}=\\mathbb{E} _ {(s,a)\\in\\mathcal{D} _ {src}}\\frac{\\hat{A}(s,a) - A(s,a)}{A(s,a)}$, where $\hat{A}$ is the estimated advantage, and $A$ is the true advantage computed by Monte Carlo. A positive $\mathcal{E}$ indicates that the advantage is overestimated, and a negative $\mathcal{E}$ means the advantage is underestimated.
>
> We conduct experiments on four datasets for evaluation: halfcheetah-morph-expert, halfcheetah-morph-medium-expert, halfcheetah-kinematic-medium and halfcheetah-kinematic-medium-replay, and we have provided the recorded advantage estimation error curves in Section 6.2 in our revision. We also present the advantage estimation error at the last step of pre-training for SQL and IQL below.
>
> ||half-morph-e|half-morph-me|half-kinematic-m|half-kinematic-mr|
> |-|-|-|-|-|
> |IQL|1.21$\pm$0.01|2.07$\pm$0.02|2.37$\pm$0.01|2.12$\pm$0.57|
> |SQL|-0.83$\pm$0.01|-0.86$\pm$0.01|-0.71$\pm$0.01|-0.49$\pm$0.01|
>
> We find that IQL tends to overestimate the advantage function, consistent with the SQL paper [1], which shows that **IQL tends to _underestimate_ the $V$-function, causing significant advantage function _overestimation_**. In contrast, SQL typically underestimates the advantage since it incorporates sparsity in $V$-function learning. Crucially, we would like to note that the absolute estimation error of SQL is **consistently smaller** than that of IQL, demonstrating that **SQL provides a more accurate advantage function for our method**.
>
> ## Concern 4: On the real-world robot experiments
>
> We appreciate this insightful suggestion. Currently, we only conduct experiments in simulation environments due to equipment limitations. But we believe it is interesting and promising to test our DVDF algorithm in real-world robot experiments, and we will explore this topic in our future work.

---

> ### Author Response · Authors · 2025-11-21
> **Reply to Reviewer qZcZ (Part 3)**
>
> ## Concern 5: More comparison and discussions with recent studies
>
> We thank the reviewer for suggesting these relevant works for comparison and discussion.
>
> PSEC [2] proposes to preserve the prior learned skills in a parametric space and adaptively composes them using a context-aware module to handle new tasks. In the cross-domain offline setting, PSEC first learns separate policies from the source domain and target domain data, which are then dynamically combined to work under the target dynamics. DmC [3] employs a kNN-based estimator as a measure of the dynamics gap, and utilizes the kNN proximity score as a guiding signal for diffusion-based data generation. Source domain samples are selected based on the proximity score and combined with the target data for training. **These prior works solely consider dynamics alignment**. **DVDF goes beyond PSEC and DmC by jointly modeling both dynamics and value alignments**.  We have added the above discussion to the related work in our revision.
>
> Moreover, as a plug-in method, **DVDF could be seamlessly integrated into both PSEC and DmC**. Specifically, DVDF could assist PSEC by selecting beneficial source domain samples to facilitate target policy learning from the limited target dataset. Similarly, DVDF could be integrated into DmC's source data selection process to enable dynamics- and value-aligned data selection. We refer to these two integrated methods as DVDF-PSEC and DVDF-DmC.
>
> We evaluate DVDF-PSEC and DVDF-DmC on four tasks (halfcheetah, hopper, walker2d, ant) under kinematic shifts, with datasets of three qualities: medium, medium-expert, and expert. All other experimental settings follow Section 6.1. The performance comparison between the base algorithms (PSEC, DmC) and their DVDF-enhanced versions (DVDF-PSEC and DVDF-DmC) is presented below.
>
>
>
>
> |Datasets|PSEC|DVDF-PSEC|DmC|DVDF-DmC|
> |-|-|-|-|-|
> |half-kine-m|**33.4$\pm$1.0**|30.1$\pm$0.8|**39.6$\pm$1.1**|33.2$\pm$0.2|
> |half-kine-me|41.6$\pm$2.3|**49.3$\pm$2.7**|47.3$\pm$5.8|**52.6$\pm$3.2**|
> |half-kine-e|52.0$\pm$4.7|**58.3$\pm$3.4**|66.9$\pm$2.4|**73.6$\pm$1.6**|
> |hopper-kine-m| 47.2$\pm$3.5|**53.2$\pm$1.5**|53.8$\pm$5.0|**65.4$\pm$2.8**|
> |hopper-kine-me|31.7$\pm$3.6|**42.0$\pm$2.4**|47.2$\pm$1.3|**56.3$\pm$2.8**|
> |hopper-kine-e|70.1$\pm$4.3|**74.2$\pm$3.4**|92.6$\pm$2.7|**98.2$\pm$1.6**|
> |walker2d-kine-m|41.0$\pm$1.9|**56.7$\pm$5.2**|48.3$\pm$4.8|**65.1$\pm$2.4**|
> |walker2d-kine-me|53.7$\pm$3.0|**58.6$\pm$3.6**|62.6$\pm$2.9|**69.2$\pm$1.2**|
> |walker2d-kine-e|75.9$\pm$1.4|**95.8$\pm$2.8**|83.0$\pm$2.6|**93.0$\pm$2.6**|
> |ant-kine-m|82.2$\pm$1.5|**87.4$\pm$1.4**|**84.1$\pm$5.5**|81.2$\pm$2.0|
> |ant-kine-me|94.6$\pm$1.7|**106.0$\pm$2.9**|107.1$\pm$3.5|**112.3$\pm$2.1**|
> |ant-kine-e|103.5$\pm$1.0|**118.4$\pm$1.5**|101.2$\pm$1.0|**116.1$\pm$2.9**|
> |total|726.9|**830.0**|833.7|**916.2**|
>
> The results show that **DVDF-PSEC outperforms PSEC on 11 out of 12 datasets, while DVDF-DmC surpasses DmC on 10 out of 12 datasets**. Furthermore, the DVDF-enhanced versions achieve a **substantially higher** total score, confirming the effectiveness and versatility of DVDF as a plug-in module. We have added the above comparisons to Appendix E.3 in our revision.
>
> [1] Offline RL with No OOD Actions: In-Sample Learning via Implicit Value Regularization. ICLR 2023
>
> [2] Skill expansion and composition in parameter space. ICLR 2025
>
> [3] DmC: Nearest Neighbor Guidance Diffusion Model for Offline Cross-domain Reinforcement Learning. ECAI 2025

---

> > ### Author Response · Authors · 2025-11-27
> >
> > Dear Reviewer qZcZ, we are grateful for your insightful review. We would like to kindly check if our rebuttal addresses your concerns. Please let us know if there is anything unclear!

---

### Official Review · Reviewer_XeL8 · 2025-10-31

**Soundness:** 3
**Presentation:** 3
**Contribution:** 3
**Rating:** 6
**Confidence:** 4

**Summary:**

This paper studies cross-domain offline RL. They showed that solely focusing on dynamics alignment as existing works do is not efficient, and illustrated the importance of value alignment. Theoretical analysis derives an upper bound on cross-domain sub-optimality, showing the necessity of considering both alignments. They presented a method (DVDF) that combine both dynamics and value alignment in the policy, and empirically, DVDF improves performance over prior filtering methods.

**Strengths:**

The paper presents a solid theoretical framework that decomposes the sub-optimality gap into dynamics misalignment and value misalignment terms. To the best of my knowledge, this idea is novel, and the analysis is mathematically sound. The propositions and proofs extend previous results to a more general setting. Overall, the paper is clearly written and well organized.

**Weaknesses:**

1. The theoretical decomposition is elegant; however, the proposed algorithm does not appear to directly optimize or approximate the theoretical quantities introduced.
2. While the theory claims to address both dynamics and value misalignment across domains, the experiments are conducted on fixed D4RL datasets with nearly identical dynamics. As a result, the method effectively filters samples based on behavioral or value similarity rather than demonstrating adaptation across genuinely different MDPs. Prior works [1, 2] introduced synthetic dynamics perturbations to evaluate such cross-domain robustness.
3. The experimental design is relatively simple, and the ablations for each component are limited to only one or two examples. A more systematic empirical analysis would strengthen the paper’s claims.

> [1] Cross-Domain Offline Policy Adaptation with Optimal Transport and Dataset Constraint (Lyu et al., 2025)
> [2] Contrastive Representation for Data Filtering in Cross-Domain Offline Reinforcement Learning (Wen et al., 2024)

**Questions:**

1. Can you clarify what specific notion of “value alignment” the paper refers to, and why this concept is theoretically distinct from existing objectives such as minimizing policy or Q-function discrepancy?
2. Equation 6 (overload notations): the $s'\in s_{tar}'\cup s_{src}'$ term looks weird. Why use the same small s to denote if s’ is a sampled state and the latter are state sets?
3. Section 5.2, line 305: g_{\xi%} is not explicitly defined. Is it the the top $\xi$-quantile of batch source sample set?
4. Table 2: the notation of data selection ratio may be inconsistent: l%, $\xi$.

---

> ### Author Response · Authors · 2025-11-21
> **Reply to Reviewer XeL8 (Part 1)**
>
> We thank the reviewer for their thoughtful feedback and for acknowledging the novelty and contribution of our work. Our point-to-point responses to the major concerns are provided below. We hope these responses could address the reveiwer's concerns.
>
> ## Concern 1: DVDF does not appear to directly optimize the introduced theoretical quantities
>
> Our main theoretical result (Proposition 4.1) establishes that the sub-optimality gap on the target domain is upper-bounded by value misalignment and dynamics misalignment. However, directly optimizing this bound is challenging due to the lack of a differentiable objective. Instead, we adopt a data filtering approach, inspired by recent advances like IGDF and OTDF, to **implicitly** optimize the bound by filtering out source domain samples that could violate it.
> Specifically, our method realizes the two alignment terms in practice as follows:
> - **Value Alignment:** As shown in Proposition 5.1, value alignment can be approximated by the in-sample optimal advantage value. Accordingly, we pre-train an advantage function and select source samples with high advantage values.
>
> - **Dynamics Alignment:** We integrate the metrics used in IGDF and OTDF to measure dynamics alignment and select samples with higher alignment scores.
>
> These two criteria are balanced by a coefficient $\lambda$, which **exactly matches** the weighted summation form of our theoretical bound in Proposition 4.1.
>
> In summary, while direct optimization is infeasible, we **implicitly tighten the sub-optimality gap through data filtering**.
>
> ## Concern 2: The experiments are conducted on D4RL datasets with nearly identical dynamics.
>
> We thank the reviewer for this important point. Our experiments adopt the same settings as in IGDF and OTDF by introducing synthetic dynamics perturbations, namely kinematic and morphology shifts, to the source domain through modifications to the .xml files, as detailed in Appendices C.2 and C.3. Therefore, the dynamics between the source and target domains are not identical but **differ significantly**, ensuring that both dynamics and value alignment play a role in our data filtering method.
> To further address the reviewer's concern and strengthen our validation, we have conducted additional experiments with another form of dynamics shift: gravity shift, as also used in IGDF and OTDF. Following OTDF, we reduce the gravity in the source domain to half:
> ```xml
> # gravity
> <option gravity="0 0 -4.905" timestep="0.01"/>
> ```
>
> We apply this gravity shift to four tasks (halfcheetah, hopper, walker2d, ant) and collect datasets of three data qualities (medium, medium-expert, expert), following the same data collection procedure as in Section 6.1. All other experimental settings remain unchanged. We compare DVDF against IGDF and OTDF, with experimental results presented below.
>
> **Gravity shift**
> |Dataset|IGDF|DVDF-IGDF|OTDF|DVDF-OTDF|
> |-|-|-|-|-|
> |half-m|43.4$\pm$0.1 | **45.7$\pm$0.1** |44.8$\pm$0.1 | **47.1$\pm$0.2** |
> |half-me| 60.4$\pm$5.8 | **73.6$\pm$6.0** | **51.3$\pm$2.6** | 46.7$\pm$0.5 |
> |half-e| 52.2$\pm$7.8 | **67.0$\pm$4.3** | 70.8$\pm$6.0 | **93.4$\pm$ 8.6**|
> |hopper-m| 47.1$\pm$3.3 | **51.8$\pm$2.1** | 46.1$\pm$3.0 | **57.6$\pm$4.8** |
> |hopper-me| 17.7$\pm$ 1.0 | **54.6$\pm$3.4** | 57.2$\pm$6.3 | **67.4$\pm$ 4.8** |
> |hopper-e| 72.3$\pm$4.8 | **95.6$\pm$ 3.6** | 100.5$\pm$1.2 | **114.6$\pm$2.5** |
> |walker2d-m| 49.4$\pm$2.0 | **60.0$\pm$1.1** | 55.8$\pm$3.9 | **73.7$\pm$2.6** |
> |walker2d-me| 74.2$\pm$3.3 | **97.6$\pm$2.4** | 65.8$\pm$4.7 | **92.0$\pm$5.1** |
> |walker2d-e| 90.1$\pm$3.7 | **105.0$\pm$2.9** | 102.4$\pm$3.8 | **113.9$\pm$0.1** |
> |ant-m| 76.1$\pm$4.5 | **87.2$\pm$ 2.6** | 86.8$\pm$1.3 | **104.8$\pm$1.7** |
> |ant-me| 107.0$\pm$0.4 | **120.5$\pm$2.6** | 110.8$\pm$3.4 | **117.7$\pm$1.3** |
> |ant-e| 115.6$\pm$3.2 | **124.6$\pm$2.1** | 111.0$\pm$ 3.9 | **126.3$\pm$ 5.2** |
> |Total| 805.5 | **983.2** | 903.3 | **1055.2** |
>
> The results show that DVDF **consistently and significantly** outperforms both baselines across most tasks. DVDF-IGDF achieves a total score of 983.2 (a **22.1% improvement** over IGDF), and DVDF-OTDF achieves 1055.2 (a **16.8% improvement** over OTDF). These consistent improvements under gravity shift, together with the results in Section 6.1, demonstrate the effectiveness of DVDF across diverse dynamics shifts and dataset qualities.

---

> ### Author Response · Authors · 2025-11-21
> **Reply to Reviewer XeL8 (Part 2)**
>
> ## Concern 3: The experimental design is relatively simple, and ablations include limited examples.
>
> We thank the reviewer for this feedback. Our original experimental design included three main components to evaluate DVDF: (1) an evaluation of its effectiveness on the D4RL benchmark under various dynamics shifts, (2) ablation studies demonstrating the necessity of the SQL-pretrained advantage function, and (3) parameter studies on $\lambda$ and $\xi$. Following Reviewer hTDh's suggestion, we have already integrated the ablation study on value-aligned filtering into the parameter sensitivity section.
>
> To address this concern, we have now conducted additional experiments on more datasets. Specifically:
>
> - We compared DVDF's performance using SQL versus IQL pre-trained advantage functions on four datasets (halfcheetah-morph-expert, halfcheetah-morph-medium-expert, halfcheetah-kinematic-medium, and halfcheetah-kinematic-medium-replay).
>
> - To further explain the performance difference, we directly compute and compare the advantage estimation error of SQL and IQL, showing that SQL provides more accurate advantage estimation.
>
> - For the parameter sensitivity analysis, we extend the evaluation of $\lambda$ and $\xi$ to four datasets each.
>
> These additional results have been incorporated into the revised manuscript (Section 6.2 and Section 6.3). We believe they provide a more comprehensive validation and could address the reviewer's concern regarding the scope of our experimental analysis.
>
> ## Concern 4: More interpretations on value alignment and the difference from existing analysis
>
>
> We appreciate the reviewer's request for clarification. In our paper, value alignment is defined as minimizing the performance gap between the learned policy and the in-sample optimal policy on the source domain (Proposition 4.1, term (a)). This emphasizes the importance of selecting optimal source samples, in addition to those with similar dynamics.
>
> Our concept of value alignment is fundamentally different from objectives like minimizing Q discrepancy in VGDF. Specifically, VGDF's objective, $\min|V(s^\prime_{src})-V(s^\prime_{tar})|$, aims to select source samples whose value is similar to target samples to reduce error in Bellman backups. However, this is unrelated to sample optimality.​ It only quantifies dynamics discrepancy from a value perspective. This is why we state that VGDF primarily addresses dynamics mismatch. Instead, our concept of value alignment introduces a new, orthogonal dimension (sample optimality) to dynamics mismatch addressed by prior work.
>
> ## Concern 5: On the overload notation $s^\prime\in s^\prime_{tar}\cup s^\prime_{src}$
>
> We thank the reviewer for pointing out this typo and the notational overload. We have corrected it in the revision by using capital $S$ to denote state sets. The corrected notation is $s^\prime\in \\{s^\prime_{tar}\\}\cup S^\prime_{src}$.
>
>
> ## Concern 6: On the undefined $g _ {\\xi\\%}$
>
> $g _ {\\xi\\%}$ means the $\xi$-th quantile of the $g$-values among source domain samples in a batch, such that we only select top $\xi\%$ source domain samples with the highest $g$-values. This definition has been explicitly stated in the revised manuscript.
>
> ## Concern 7: On the inconsistent notation of data selection ratio
>
> We thank the reviewer for pointing this out. We have unified the notation for the data selection ratio to $\xi$ in our revision.

---

> > ### Author Response · Authors · 2025-11-28
> >
> > Dear Reviewer XeL8, thank you for your constructive review! We hope that our rebuttal and the revised manuscript can address your concerns. Please let us know if there is still anything unclear!

---

### Meta-Review · Area_Chair_4qQS · 2026-01-07

**Summary:**

The paper proposes Dynamics- and Value-aligned Data Filtering (DVDF) for cross-domain offline RL. While the authors made significant efforts during the rebuttal to add baselines (PSEC, DmC) and new dynamics shift settings (Gravity Shift), the consensus leans towards Reject. This decision is primarily informed by the following outstanding concerns:

Hyperparameter Sensitivity and Robustness: Reviewer hTDh raised critical concerns regarding the sensitivity of the tradeoff parameter $\lambda$. The rebuttal analysis confirmed that the optimal $\lambda$ varies significantly across tasks, and the authors acknowledged that it cannot be determined without task-specific tuning. In an offline RL setting where online tuning is not feasible, this fragility limits the method's practical utility.

High Variance: Reviewer hTDh noted that the results exhibit high variance (e.g., standard deviations overlapping significantly), making the claimed improvements unconvincing.

Theory-Practice Gap: Reviewer XeL8 pointed out that while the theoretical decomposition of the sub-optimality gap is elegant, the proposed algorithm does not directly optimize these quantities, serving instead as a heuristic filtering mechanism. This disconnect weakens the theoretical contribution.

**Reviewer Concerns:**

The initial reviews were lukewarm (6, 4, 4), highlighting significant concerns regarding the experimental setup, baseline comparisons, and the justification for the chosen advantage estimation method.

Addressed Concerns:

Missing Baselines: Reviewer qZcZ rightly pointed out the lack of comparison with recent state-of-the-art methods like PSEC and DmC. The authors did a commendable job during the rebuttal by implementing DVDF as a plug-in for these methods, showing performance gains on the majority of datasets.

Dynamics Shift Validity: Reviewer XeL8 critiqued the original experiments for using D4RL datasets where dynamics are nearly identical. The authors responded by adding "Gravity Shift" and "Kinematic Shift" (broken joints) experiments, which creates a more realistic cross-domain setting.

Advantage Estimation Choice: There was skepticism from Reviewers qZcZ and hTDh regarding the choice of SQL over IQL. The authors provided empirical evidence showing SQL has lower absolute advantage estimation error compared to IQL, which is a sufficient justification for this design choice.

Outstanding Concerns:

Hyperparameter Sensitivity & Robustness: This is the most significant remaining issue. Reviewer hTDh raised concerns about the sensitivity of the weighting parameter $\lambda$. The authors' new sensitivity analysis reveals that performance is highly volatile depending on $\lambda$ (e.g., on hopper-kinematic-me, scores range from 14.6 to 63.4). While the authors claim a fixed $\lambda=0.7$ works "generally well," they openly admit that "optimal $\lambda$ cannot be determined without some task-specific tuning". In offline RL, where online tuning is impossible, this fragility is a major drawback.

Theory-Practice Gap: Reviewer XeL8 noted that while the theoretical decomposition is elegant, the algorithm does not directly optimize these quantities. It remains a heuristic filtering approach inspired by the bound rather than a direct minimization of the sub-optimality gap. The rebuttal clarified this is an "implicit" optimization, but the disconnect remains.

Performance Variance: Reviewer hTDh noted high variance in the results. Even with the new experiments, the method's reliance on pre-trained advantage functions (which themselves have estimation errors) adds layers of uncertainty that the current evaluation doesn't fully mitigate.

**Reviewer Scores:**

Reviewer XeL8: 6 (Marginally above threshold). Appreciates the theory but notes the method is effectively a heuristic filter.

Reviewer qZcZ: 4 (Marginally below threshold). Acknowledged the new baselines but remains lukewarm; the "real-world" concern was addressed by staying in sim, but the overall novelty is limited.

Reviewer hTDh: 4 (Marginally below threshold). Remains concerned about the high variance and the practical fragility of the hyperparameter $\lambda$.

---

### Decision · Program_Chairs · 2026-01-26

Reject